# Exogenous Melatonin Reinforces Photosynthesis, Antioxidant Defense and Gene Expression to Ameliorate Na_2_CO_3_ Stress in Maize

**DOI:** 10.3390/plants13202844

**Published:** 2024-10-11

**Authors:** Guoxiang Qi, Xiaoqiang Zhao, Fuqiang He, Siqi Sun, Zhenzhen Shi, Yining Niu

**Affiliations:** State Key Laboratory of Aridland Crop Science, College of Agronomy, Gansu Agricultural University, Lanzhou 730070, China; qigx1321@163.com (G.Q.); hefq6125@163.com (F.H.); 15045240973@163.com (S.S.); shizz@gsau.edu.cn (Z.S.); niuyn@gsau.edu.cn (Y.N.)

**Keywords:** maize seedlings, Na_2_CO_3_ stress, melatonin, ethyl methane sulfonate (EMS) mutant, antioxidant system, photosynthesis, quantitative real-time PCR

## Abstract

Salt stress can seriously affect the growth and development of maize (*Zea mays* L.), resulting in a great yield loss. Melatonin (MT), an indole hormone, is a potential enhancer of plant tolerance against salt stress. However, the complex mechanisms of MT application in enhancing maize salt tolerance are still unclear. Herein, three-leaf seedlings of salt-susceptible P138 and its salt-resistant ethyl methane sulfonate (EMS)-104 mutant were cultured with or without 150 μM MT application under 0 and 100 mM Na_2_CO_3_ treatments for seven days, to systematically explore the response mechanisms of exogenous MT in improving the salt tolerance of maize. The results showed that salt stress triggered an escalation in reactive oxygen species production, enhanced multiple antioxidant enzymes’ activities, impaired cellular membrane permeability, inhibited photosynthetic pigment accumulation, and ultimately undermined the vigor and photosynthetic prowess of the seedlings. While suitable MT application counteracted the detrimental impacts of Na_2_CO_3_ on seedlings’ growth and photosynthetic capacity, the seedling length and net photosynthetic rate of P138 and EMS-104 were increased by 5.5% and 18.7%, and 12.7% and 54.5%, respectively. Quantitative real-time PCR (qRT-PCR) analysis further showed that MT application activated the expression levels of antioxidant enzyme-related genes (*Zm00001d025106*, *Zm00001d031908*, *Zm00001d027511*, and *Zm00001d040364*) and pigment biosynthesis-related genes (*Zm00001d011819* and *Zm00001d017766*) in both maize seedlings under Na_2_CO_3_ stress; they then formed a complex interaction network of gene expression, multiple physiological metabolisms, and phenotype changes to influence the salt tolerance of maize seedlings under MT or Na_2_CO_3_ stress. To sum up, these observations underscore that 150 μM MT can alleviate salt injury of maize seedlings, which may provide new insights for further investigating MT regulation mechanisms to enhance maize seedlings’ salt resistance.

## 1. Introduction

Salt stress represents a significant constraint on crop yields worldwide, jeopardizing global food security [1]. Globally, saline–alkali soil covers about 9.54 × 10^8^ ha, with China alone possessing 9.91 × 10^7^ ha [2,3,4]. By 2050, it is expected that about 50% of agricultural land worldwide will be affected by salinity [4]. The salinization problem is caused by a variety of factors, including little rainfall, high evaporation rates, improper irrigation, and excessive fertilizer use [5,6]. Due to high salt levels severely hindering the absorption of essential nutrients and water in different crop species, their growth and development are inhibited and toxic ion accumulation is aggravated [7].

Maize (*Zea mays* L.), an important cereal crop, plays a pivotal role in providing food, feed, and bioenergy. Generally, maize is extremely sensitive to salt stress during its whole growth period, especially the seedling stage [8,9]. Salt stress influences the activities of multiple photosynthetic enzymes in maize, such as adenosine triphosphate synthetase (ATPase), phosphoenol-pyruvate carboxylase (PEPcase), and ribulose-1,5-bisphosphate carboxylase (RuBPase), which further inhibits CO_2_ fixation and photosynthesis efficiency [10,11,12], ultimately contributing to yield losses of 20–50% [13,14]. At the same time, excess free radicals and reactive oxygen species (ROS) elevate lipid peroxidation and lipoxygenase acidity, which further decomposes phospholipids of cellular membranes in maize seedlings under salt stress [15]. In addition, stressed maize seedlings cannot efficiently absorb and assimilate nutrients from the soil, leading to nutritional imbalances [16].

Melatonin (MT), an indole compound, shares structural similarities with indole-3-acetic acid (IAA) and exhibits a characteristic “low concentration promotion, high concentration inhibition” behavior [17]. Previous studies have reported that MT could benefit growth, biomass accumulation, flowering, and grain formation in different plants [18,19,20]. Meanwhile, MT has been shown to alleviate salt stress in various crop species, including rice (*Oryza sativa* L.), soybean (*Glycine max* L.), cucumber (*Cucumissativus* L.), and sunflower (*Helianthus annuus* L.) [21,22,23,24]. In particular, MT effectively inhibited chlorophyll degradation while ensuring the normal production of photosynthetic pigments under salt stress [25]. The application of 100 μM MT enhances the net photosynthetic rate (Pn) in tomato (*Solanum lycopersicum* L.) seedlings under salt stress, which exhibit better growth [26]. Likewise, 100 μM MT application increases stomatal conductance (Gs) and intercellular CO_2_ accumulation (Ci) of soybean leaves under drought stress, thereby enhancing the drought resistance of soybean plants [17]. In addition, MT also serves as an effective inducer and direct eliminator of excessive ROS inside cells, thus modulating oxidative damage resulting from environmental stresses [27]. However, its role in alleviating Na_2_CO_3_ stress in maize seedlings is not well known.

Considering the crucial importance of MT in boosting plant resistance, it is of great significance to reveal the mechanism of exogenous MT application in the mitigation of salt stress in maize seedlings. In this investigation, the salt-susceptible inbred maize line P138 and its salt-resistant EMS-104 (from a library of ethyl methane sulfonate (EMS)-induced mutants) mutant were used as materials. Their seedlings at the three-leaf stage were treated with or without 150 μM MT application under 0 and 100 mM Na_2_CO_3_ treatments for seven days to analyze seedling growth status, photosynthetic pigment accumulation, photosynthetic ability, ROS levels, and antioxidant enzyme activities; meanwhile, the expression levels of six candidate genes responsible for antioxidant enzymes and photosynthesis in both genotypes seedlings under all treatments were also quantitatively analyzed by quantitative real-time PCR (qRT-PCR). Thereby, this study will help us determine the mechanism by which MT alleviates the impact of salt stress on maize seedlings, and gather insights into the mechanisms of salt tolerance in maize.

## 2. Results

### 2.1. Combined Analysis of All Traits

Multiple factorial analysis of variance (ANOVA) proved significant for the tested variables (including maize genotype (G), MT (M) application, and Na_2_CO_3_ (N) stress) and their physiological and molecular interactions with twenty-five tested traits including four growth parameters, eight photosynthetic performances, one membrane characteristic, one ROS level, four antioxidant enzymes’ activities, one cell osmotic pressure, and six stomatal morphology traits in two maize genotype seedlings under different treatments (Figure 1, Figure 2, Figure 3, Figure 4, Figure 5 and Figure 6; Appendix A). The findings demonstrated that resistance to salt stress was controlled by maize’s own genetic constitution, exogenous MT application, Na_2_CO_3_ stress, and their interaction effects. Our results also showed that the genotypes, MT application, and Na_2_CO_3_ stress imposed higher influences on these traits than their interactions.

### 2.2. Effect of MT on Changes in Growth Phenotypes of Maize Seedlings under Na_2_CO_3_ Stress

The growth phenotypes can directly reflect the changes in plants under different treatments. In this study, under N (100 mM Na_2_CO_3_ + 0 μM MT) stress, P138 and EMS-104 seedlings showed significant decreases in seedling weight (SW), root weight (RW), seedling length (SL), and leaf area (LA) compared to CK (0 mM Na_2_CO_3_ + 0 μM MT) treatment, with 29.0% and 27.2%, 32.8% and 29.9%, 26.5% and 12.8%, and 26.4% and 24.9% decreases, respectively (*p* < 0.05) (Figure 1A–D). This phenomenon suggested that 100 mM Na_2_CO_3_ stress can significantly inhibit maize seedling growth. In the M (0 mM Na_2_CO_3_ + 150 μM MT) treatment, only P138 seedlings had higher seedling weight and root weight, which increased by 11.7% and 6.6%, respectively (Figure 1A,B). When both maize genotypes were treated with MN (100 mM Na_2_CO_3_ + 150 μM MT), P138 root weight, seedling length, and leaf area increased by 27.9%, 18.8%, and 26.8%, respectively (Figure 1B–D). The EMS-104 seedlings showed significantly increased seedling weight and leaf area by 13.8% and 18.0% under MN treatment (Figure 1A,D). These results showed that 150 μM MT application clearly alleviates the inhibition of maize seedling growth when they are growing in a Na_2_CO_3_ environment. Overall, the growth of mutant EMS-104 seedlings was better than the wild P138 seedlings under all treatments (Figure 1).

**Figure 1 plants-13-02844-f001:**
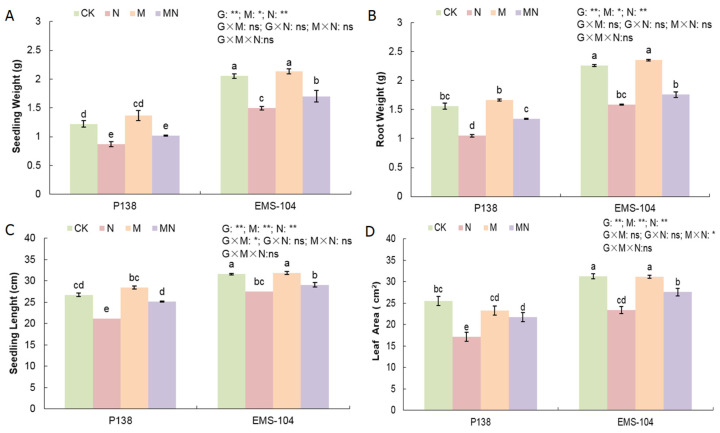
Changes in seedling weight (**A**), root weight (**B**), seedling length (**C**), and leaf area (**D**) of P138 and EMS-104 seedlings under different treatments, including 0 mM Na_2_CO_3_ + 0 μM melatonin (MT) treatment (CK); 100 mM Na_2_CO_3_ + 0 μM MT treatment (N); 0 mM Na_2_CO_3_ + 150 μM MT treatment (M); and 100 mM Na_2_CO_3_ + 150 μM MT treatment (MN). Different lowercase letters indicate significant differences at the *p* < 0.05 level. ns: multiple factorial analysis of variance (ANOVA) was nonsignificant at the *p* > 0.05 level; *: multiple factorial ANOVA was significant at the *p* ≤ 0.05 level; **: multiple factorial ANOVA was significant at the *p* ≤ 0.01 level.

### 2.3. Effects of MT on H_2_O_2_ Content and Antioxidant Enzyme Activities of Maize Seedlings under Na_2_CO_3_ Stress

The maintenance of reactive oxygen species (ROS) homeostasis is pivotal for normal growth, development, and stress tolerance in plants [28]. Regarding changes in the content of hydrogen peroxide (H_2_O_2_), an important ROS, it was significantly increased by 70.4% and 79.5% in P138 and EMS-104 seedlings under N stress compared to the control (*p* < 0.05) (Figure 2E). The findings suggested that the presence of high Na_2_CO_3_ concentration stimulates high H_2_O_2_ production in maize seedlings. Under M treatment, the H_2_O_2_ content in P138 and EMS-104 seedlings decreased by 14.7% and 11.9%, respectively (*p* < 0.05) (Figure 2E). Compared to N stress, the H_2_O_2_ content in P138 and EMS-104 seedlings decreased further by 32.2% and 30.5% under MN treatment, respectively (Figure 2E). The findings indicated that MT possibly inhibits the generation of H_2_O_2_ or scavenges its deposition (Figure 2E).

The antioxidant enzyme system plays an important role in protecting plant cells from stress [29]. In this study, compared with the control, N stress significantly increased the activities of superoxide dismutase (SOD), peroxidase (POD), catalase (CAT), and ascorbate peroxidase (APX) in P138 and EMS-104 seedlings (*p* < 0.05), they increased by 76.5% and 94.9%, 17.5% and 25.7%, 187.8% and 216.9%, and 144.2% and 84.8%, respectively (Figure 2A–D). The results showed that Na_2_CO_3_ stress enhances antioxidant enzyme activities to scavenge excess ROS in maize. Moreover, the SOD activity of P138 seedlings significantly increased by 24.9%, while the APX activity clearly decreased by 19.2% under M treatment; similarly, SOD activity increased significantly by 55.8%; however, CAT activity decreased clearly by 84.9% in EMS-104 seedlings under M treatment (Figure 2A,D). Additionally, when both seedlings of P138 and EMS-104 were exposed to MN treatment, their four enzymes’ activities increased by 7.9% and 3.4%, 8.9% and 6.9%, 22.1% and 10.7%, and 30.7% and 27.3% compared to N stress, respectively (Figure 2A–D).

**Figure 2 plants-13-02844-f002:**
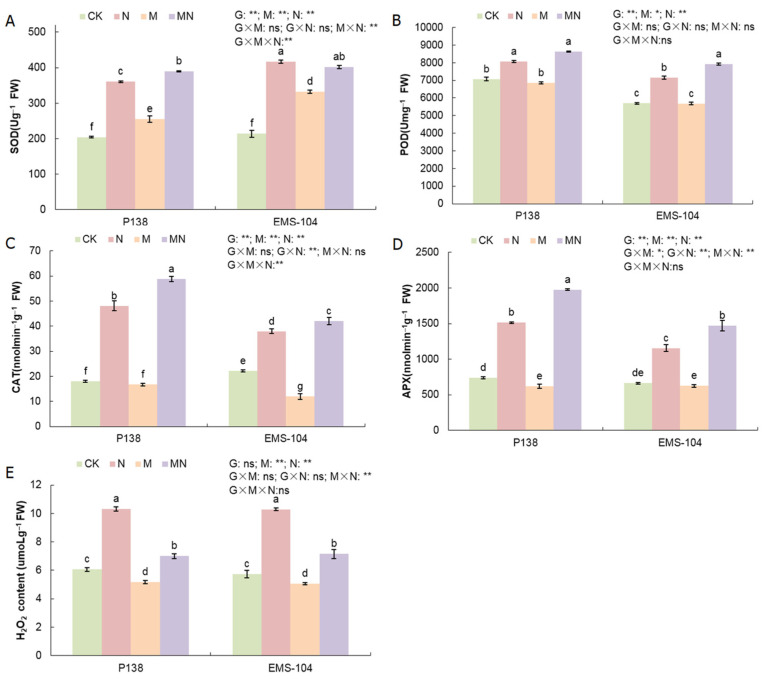
Changes in superoxide dismutase (SOD) activity (**A**), peroxidase (POD) activity (**B**), catalase (CAT) activity (**C**), ascorbate peroxidase (APX) activity (**D**), and H_2_O_2_ content (**E**) of P138 and EMS-104 seedlings under different treatments, including 0 mM Na_2_CO_3_ + 0 μM melatonin (MT) treatment (CK); 100 mM Na_2_CO_3_ + 0 μM MT treatment (N); 0 mM Na_2_CO_3_ + 150 μM MT treatment (M); and 100 mM Na_2_CO_3_ + 150 μM MT treatment (MN). Different lowercase letters indicate significant differences at the *p* < 0.05 level. ns: multiple factorial analysis of variance (ANOVA) was nonsignificant at the *p* > 0.05 level; *: multiple factorial ANOVA was significant at the *p* ≤ 0.05 level; **: multiple factorial ANOVA was significant at the *p* ≤ 0.01 level.

### 2.4. Effect of MT on Relative Electrical Conductivity and Relative Water Content of Maize Seedlings under Na_2_CO_3_ Stress

To evaluate the cell status of maize seedlings under Na_2_CO_3_ stress, the relative electrical conductivity (REC) and relative water content (RWC) in the leaves of two maize genotypes under different treatments were measured. Compared to CK, both P138 and EMS-104 leaves had significantly increased REC, with 33.3% and 20.5%, and had a significant decrease in RWC, with 4.2% and 1.8% under N treatment (Figure 3A,B). The results showed that Na_2_CO_3_ stress could aggravate cell membrane peroxidation to cause a significant decrease in RWC in maize. In the MN treatment, P138 leaves showed a lower REC than that under N treatment (9.4%; *p* < 0.05), while the RWC in P138 leaves increased by 2.5% (Figure 3A,B).

**Figure 3 plants-13-02844-f003:**
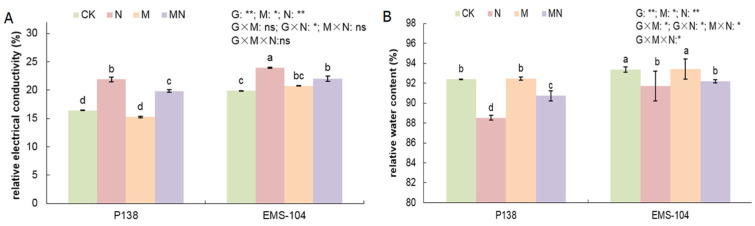
Changes in relative electrical conductivity (REC (**A**)), and relative water content (RWC (**B**)) of P138 and EMS-104 seedlings under different treatments, including 0 mM Na_2_CO_3_ + 0 μM melatonin (MT) treatment (CK); 100 mM Na_2_CO_3_ + 0 μM MT treatment (N); 0 mM Na_2_CO_3_ + 150 μM MT treatment (M); and 100 mM Na_2_CO_3_ + 150 μM MT treatment (MN). Different lowercase letters indicate significant differences at the *p* < 0.05 level. ns: multiple factorial analysis of variance (ANOVA) was nonsignificant at the *p* > 0.05 level; *: multiple factorial ANOVA was significant at the *p* ≤ 0.05 level; **: multiple factorial ANOVA was significant at the *p* ≤ 0.01 level.

### 2.5. Effects of MT on Photosynthetic Characteristics of Maize Seedlings under Na_2_CO_3_ Stress

Compared to CK treatment, N stress led to significant decreases in Pn, Gs, and transpiration rate (Tr) of both P138 and EMS-104 genotypes, with 57.6% and 31.9%, 56.6% and 30.1%, and 72.3% and 56.8%, respectively, while the Ci increased significantly by 79.3% and 142.7% (*p* < 0.05) (Figure 4A–D). The M treatment had lower values of Pn, Gs, and Ci for P138 material, while its Tr increased by 11.3%; EMS-104 seedlings showed increased Gs, Ci, and Tr by 4.6%, 23.1%, and 1.4% under M treatment (Figure 4A–D). Under MN treatment, P138 and EMS-104 seedlings exhibited increases in Pn, Gs, and Tr by 54.6% and 12.8%, 75.1% and 29.5%, and 146.9% and 60.1%, respectively, while their Ci decreased by 22.0% and 23.5%, respectively (Figure 4A–D).

**Figure 4 plants-13-02844-f004:**
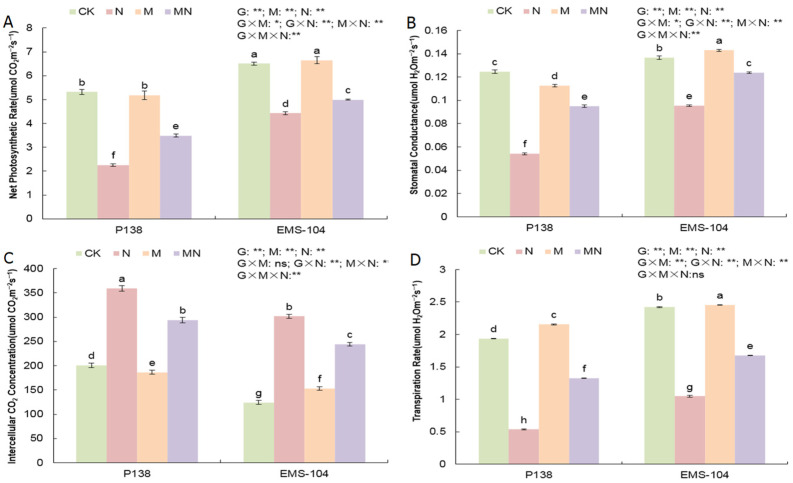
Changes in net photosynthetic rate (**A**), stomatal conductance (**B**), intercellular CO_2_ concentration (**C**), and transpiration rate (**D**) of P138 and EMS-104 seedlings under different treatments, including 0 mM Na_2_CO_3_ + 0 μM melatonin (MT) treatment (CK); 100 mM Na_2_CO_3_ + 0 μM MT treatment (N); 0 mM Na_2_CO_3_ + 150 μM MT treatment (M); and 100 mM Na_2_CO_3_ + 150 μM MT treatment (MN). Different lowercase letters indicate significant differences at the *p* < 0.05 level. ns: multiple factorial analysis of variance (ANOVA) was nonsignificant at the *p* > 0.05 level; *: multiple factorial ANOVA was significant at the *p* ≤ 0.05 level; **: multiple factorial ANOVA was significant at the *p* ≤ 0.01 level.

### 2.6. Effect of MT on Photosynthetic Pigments in Maize Seedlings under Na_2_CO_3_ Stress

The contents of multiple photosynthetic pigments of two maize genotypes also showed the same changes under different treatments. In comparison to CK, the N treatment significantly decreased chlorophyll a (Chl a) content, chlorophyll b (Chl b) content, carotenoid (Car) content, and the chlorophyll a/b ratio (Chl a/b) in all maize materials, by 29.4% and 28.7%, 14.1% and 18.1%, 53.7% and 64.1%, and 23.9% and 25.6%, respectively (*p* < 0.05) (Figure 5A–D). The results showed that Na_2_CO_3_ stress could damage chloroplasts’ structure and inhibit photosynthetic pigment synthesis in maize, and the influence of Chl a content was larger than that of Chl b content. Conversely, the M treatment led to significant increases in these parameters for P138 and EMS-104 seedlings, with 22.0% and 5.0%, 5.9% and 6.1%, 15.5% and 3.8%, and 16.3% and 5.4%, respectively (*p* < 0.05) (Figure 5A–D). Under MN treatment, P138 and EMS-104 seedlings had increases in these traits by 29.1% and 17.4%, 9.5% and 16.5%, 60.4% and 46.7%, and 21.2% and 17.1%, respectively (*p* < 0.05) (Figure 5A–D). The results thus showed that 150 μM MT application might protect chloroplasts’ structure and increase multiple photosynthetic pigments’ accumulation in maize seedlings under Na_2_CO_3_ stress.

**Figure 5 plants-13-02844-f005:**
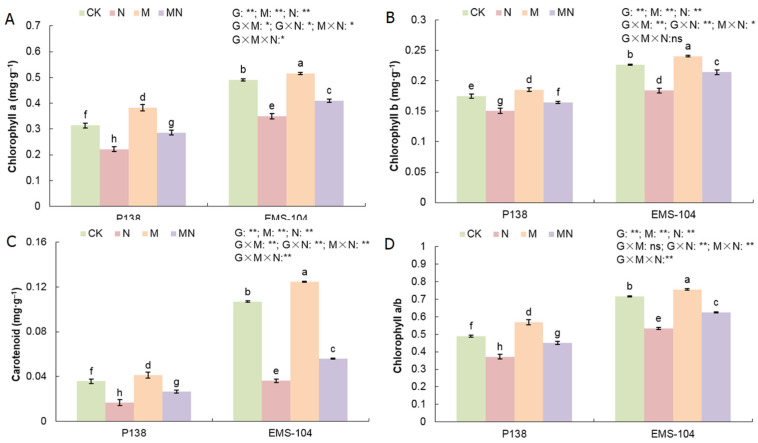
Changes in chlorophyll a (Chl a (**A**)), chlorophyll b (Chl b (**B**)), carotenoids (Car; (**C**)), and chlorophyll a/b (Chl a/b (**D**)) of P138 and EMS-104 seedlings under different treatments, including 0 mM Na_2_CO_3_ + 0 μM melatonin (MT) treatment (CK); 100 mM Na_2_CO_3_ + 0 μM MT treatment (N); 0 mM Na_2_CO_3_ + 150 μM MT treatment (M); and 100 mM Na_2_CO_3_ + 150 μM MT treatment (MN). Different lowercase letters indicate significant differences at the *p* < 0.05 level. ns: multiple factorial analysis of variance (ANOVA) was nonsignificant at the *p* > 0.05 level; *: multiple factorial ANOVA was significant at the *p* ≤ 0.05 level; **: multiple factorial ANOVA was significant at the *p* ≤ 0.01 level.

### 2.7. Effects of MT on Stomata of Maize Seedlings under Na_2_CO_3_ Stress

To verify the cause of Pn decrease after Na_2_CO_3_ stress, the stomatal morphology of the leaves in maize seedlings was observed. Under N stress, the paraxial surface stomatal length (PSL), paraxial surface stomatal width (PSW), and paraxial surface stomatal area (PSA) of P138 and EMS-104 experienced a significant decrease by 21.0% and 9.0%, 13.7% and 11.6%, and 28.0% and 16.6%, respectively (*p* < 0.05) (Figure 6A,C,E). The same phenomenon was also observed in dorsal surface stomatal length (DSL), dorsal surface stomatal width (DSW), and dorsal surface stomatal area (DSA) in these maize materials, which decreased by 28.9% and 6.9%, 29.2% and 27.2%, and 30.6% and 27.6%, respectively (*p* < 0.05) (Figure 6B,D,F). We thus speculated that the decrease in Pn is correlated with the changes in paraxial/dorsal surface stomata in maize seedling leaves under Na_2_CO_3_ stress. Under M treatment, there were no clear changes in the paraxial surface stomatal morphology of P138 leaves; meanwhile, the paraxial surface stomatal morphology parameters, including PSL, PSW, and PSA, in EMS-104 leaves were increased by 11.0%, 4.2%, and 8.9%, respectively (*p* < 0.05) (Figure 6A,C,E). Interestingly, for the dorsal surface stomata, the DSL, DSW, and DSA in P138 leaves showed significant increases under M treatment by 22.9%, 8.4%, and 12.5%, respectively; except for DSW, other dorsal surface stomatal morphology parameters in EMS-104 leaves had significant increases (*p* < 0.05) (Figure 6B,D,F). When P138 and EMS-104 seedlings were cultured in MN treatment, their three paraxial surface stomatal morphology parameters were increased by 10.7% and 3.4%, 12.2% and 9.9%, and 17.3% and 8.3%, respectively; similarly, their three dorsal surface stomatal morphology parameters were also increased by 43.3% and 10.4%, 15.4% and 24.8%, and 25.6% and 31.0%, respectively (*p* < 0.05) (Figure 6A–F). Therefore, these results showed that MT application could improve multiple paraxial/dorsal surface stomatal morphology in maize leaves under Na_2_CO_3_ stress.

**Figure 6 plants-13-02844-f006:**
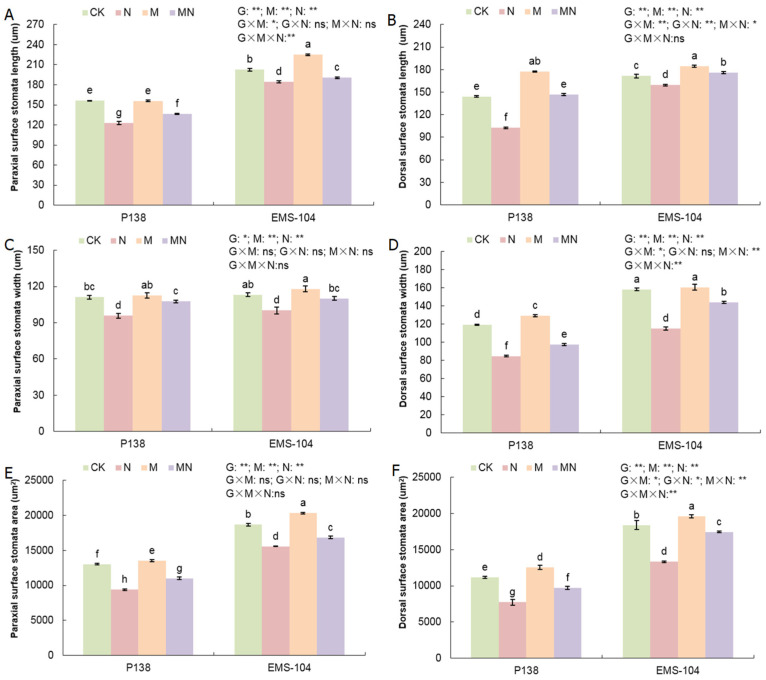
Changes in paraxial surface stomatal length (**A**), dorsal surface stomatal length (**B**), paraxial surface stomatal width (**C**), dorsal surface stomatal width (**D**), paraxial surface stomatal area (**E**), and dorsal surface stomatal area (**F**) of P138 and EMS-104 seedlings under different treatments, including 0 mM Na_2_CO_3_ + 0 μM melatonin (MT) treatment (CK); 100 mM Na_2_CO_3_ + 0 μM MT treatment (N); 0 mM Na_2_CO_3_ + 150 μM MT treatment (M); and 100 mM Na_2_CO_3_ + 150 μM MT treatment (MN). Different lowercase letters indicate significant differences at the *p* < 0.05 level. ns: multiple factorial analysis of variance (ANOVA) was nonsignificant at the *p* > 0.05 level; *: multiple factorial ANOVA was significant at the *p* ≤ 0.05 level; **: multiple factorial ANOVA was significant at the *p* ≤ 0.01 level.

### 2.8. Correlation Relationships among 25 Traits in Two Maize Genotypes under All Treatments

In this study, a Pearson correlation analysis among 25 tested traits in two maize genotypes under all treatments was further conducted to reveal the internal relationships among all traits under Na_2_CO_3_ stress and MT stimulation. The results showed that the REC was significantly negatively correlated with Tr, PSW, and SOD activity; in contrast, it was significantly positively correlated with Ci, CAT activity, APX activity, and H_2_O_2_ levels. The Ci was negatively correlated with Tr, PSL, DSL, PSW, DSW, PSA, DSA, Chl a content, Chl b content, Car content, Chl a/b, and SOD activity; however, it was significantly positively correlated with POD activity, CAT activity, APX activity, and H_2_O_2_ content. The Pn was significantly positively correlated with Gs, Tr, PSL, DSL, PSW, DSW, PSA, DSA, Chl a content, Chl b content, Car content, Chl a/b, and SOD activity; it was significantly negatively correlated with POD activity, CAT activity, APX activity, and H_2_O_2_ content. The H_2_O_2_ content was significantly negatively correlated with SW, RW, SL, LA, RWC, Pn, Gs, Tr, PSL, DSL, PSW, DSW, PSA, DSA, Chl a content, Chl b content, Car content, Chl a/b, and SOD activity. Conversely, there was a significant positive correlation with REC, Ci, SOD activity, CAT activity, and APX activity (Figure 7). These results suggested that multiple traits synergistically influence salt tolerance in maize under salt stress or MT treatment.

### 2.9. Expression Analysis of Genes Involved in Photosynthesis and Antioxidant Homeostasis under Na_2_CO_3_ Stress

To further verify the regulatory mechanism of Na_2_CO_3_ stress in maize seedlings, the differential expression levels of six candidate genes involved in photosynthesis and antioxidant homeostasis were analyzed. Compared to CK treatment, N stress significantly increased the corresponding gene expression levels for *superoxide dismutase* (*SOD*; *Zm00001d025106* and *Zm00001d031908*), *catalase 2* (*CAT2*; *Zm00001d027511*), and *peroxidase 72* (*POD72*; *Zm00001d040364*) in both P138 and EMS-104 leaves by 98.8% and 137.4%, 39.6% and 26.4%, 63.8% and 88.2%, and 15.9% and 11.4%, respectively (*p* < 0.05) (Figure 8A–D). The expression levels of two photosynthesis-related genes, i.e., *nine-cis-epoxycarotenoid dioxygenase8* (*Zm00001d017766*) and *chlorophyllide a oxygenase chloroplastic* (*Zm00001d011819*), were significantly decreased by 53.6% and 25.5%, and 49.2% and 9.8%, respectively (*p* < 0.05) (Figure 8E,F). Under M treatment, the expressions of *Zm00001d025106*, *Zm00001d011819*, and *Zm00001d017766* in P138 and EMS-104 increased by 36.8% and 53.7%, 28.3% and 45.6%, and 57.7% and 69.1%, compared to CK, and *Zm00001d031908* and *Zm00001d040364* in P138 were up-regulated by 24.7% and 11.4%; notably, the expression pattern of *Zm00001d027511* decreased by 12.1% and 46.1% (*p* < 0.05) (Figure 8A–F). Moreover, in the MN treatment, *Zm00001d025106*, *Zm00001d031908*, *Zm00001d040364*, *Zm00001d011819*, and *Zm00001d017766* in P138 and EMS-104 were significantly up-regulated by 59.5% and 33.9%, 53.0% and 36.3%, 15.2% and 16.8%, 137.4% and 69.2%, and 194.6% and 48.8% compared to N stress, respectively (*p* < 0.05) (Figure 8A–F).

## 3. Discussion

It is well known that salt stress is the major environmental factor in the soil that inhibits and negatively affects plant growth, development, and yield formation [22,30]. Fortunately, different strategies have been developed in some plant species to resist various environmental stresses. Previous studies have shown that exogenous MT application can enhance salt tolerance in rice, soybean, cucumber, and sunflower [21,22,23,24]. However, our knowledge about the mechanisms involved in MT-mediated tolerance to Na_2_CO_3_ stress in salt-susceptible P138 and its salt-resistant EMS-104 mutant seedlings still remains unclear.

Many studies clearly observed that the phenotypes of plants under salt stress exhibited unfavorable effects, especially in biomass and plant height [31,32,33]. Surprisingly, 150 μM MT increased the leaf area and biomass of cotton (*Gossypium hirsutum* L.) plants under salt stress [34]. A similar phenomenon was also found in our study, i.e., under the stress of 100 mM Na_2_CO_3_, the seedling length, leaf area, seedling weight, and root weight of P138 and theEMS-104 mutant were significantly reduced (Figure 1A–D), indicating that salt stress severely inhibits the survival or healthy growth of maize. We also found that the growth status of theEMS-104 mutant was better than its wild-type P138 (Figure 1A–D), which provided valuable genetic resources for salt tolerance breeding in future. After applying 150 μM MT, these growth phenotypes were improved significantly (Figure 1A–D). From this, we infer that MT application may improve multiple physiological processes of maize seedlings to resist Na_2_CO_3_ stress injury.

The RWC can directly reflect the water status of plants [35], and the REC is generally considered to show membrane permeability and membrane integrity under stress [36]. Zhao et al. [37] found that drought stress reduced the RWC of maize, and Wang et al. [38] found that salt stress combined with drought stress increased the REC of boxwood seedlings. Interestingly, early studies have shown that the application of MT was beneficial for maintaining intracellular turgor pressure and improving plasma ion homeostasis [38,39]. In this study, we found that the RWC of maize P138 and EMS-104 genotypes decreased significantly, while their REC was significantly increased under the stress of 100 mM Na_2_CO_3_, which was consistent with the results of previous studies [37,38]. After applying 150 μM MT, the RWC of P138 and EMS-104 seedlings was significantly increased, but the REC was significantly decreased (Figure 3A,B), indicating that 150 μM MT treatment may maintain relatively high osmotic pressure to protect cell membrane integrity in maize seedlings under 100 mM Na_2_CO_3_ stress.

Photosynthetic pigments are closely related to photosynthesis and stress resistance in plants [40,41]. Salt stress accelerated the degradation of chlorophyll and inhibited the biosynthesis of chlorophyll in maize [42]. Under drought stress, Ahmad et al. [43] discovered that chlorophyll levels were clearly increased in maize after 100 μM MT application. Regarding other exogenous hormones’ applications, similar results have also confirmed that exogenous salicylic acid (SA) increased the chlorophyll content of maize under salt stress [44]. Our study showed that 100 mM Na_2_CO_3_ stress significantly decreased the contents of Chl a, Chl b, and Car in P138 and EMS-104 seedling leaves. In contrast, 150 μM MT treatment significantly increased these photosynthetic pigments’ contents (Figure 5A–D). This is consistent with the results of previous studies [43,44]. In addition, our gene expression analysis also showed that *Zm00001d017766* and *Zm00001d011819*, two photosynthetic pigment-related genes had positive expression levels under MT application in both maize genotypes (Figure 8E,F). Therefore, these findings indicated that 150 μM MT could promote photosynthetic pigments’ gene expression to increase photosynthetic pigment accumulation under 100 mM Na_2_CO_3_ stress. In addition, some studies have shown that appropriate ROS levels are beneficial for chloroplast development and function [45]. We also found that levels of ROS, such as H_2_O_2_, were significantly decreased in P138 and EMS-104 seedlings when they were treated with 150 μM MT application under 100 mM Na_2_CO_3_ stress (Figure 2E). Thereby, lower ROS levels may benefit photosynthetic pigment production and maintain high photosynthesis rates in maize under various environmental stimulations.

The changes in stomatal morphology can positively reflect the physiological changes in plants under different environmental conditions [46]. Some studies have shown that salt stress led to stomatal closure, while exogenous silicon (Si) treatment improved the Gs of alfalfa (*Medicago sativa* L.) seedlings under salt stress [47]. Similarly, exogenous SA application improved the Gs of barley under salt stress [48]. We also found that the application of 150 μM MT improved the paraxial/dorsal surface stomatal morphology under 100 mM Na_2_CO_3_ stress in the current study (Figure 6A–F).

Photosynthesis is the fundament for plants’ survival, growth, reproduction, and resilience to environmental stresses [49,50]. Salt stress reduced the light energy absorption and conversion efficiency of leaves [51]. Previous studies have shown that exogenous MT protected the structure of chloroplasts and photosynthetic organs, induced stomatal reopening, sustained high levels of chlorophyll, and enhanced leaves’ ability to capture light energy; this action thus alleviated the inhibition of photosynthesis caused by NaCl stress in alfalfa leaves [52,53,54,55]. Similar studies also found that MT stimulation significantly enhanced the photosynthetic efficiency in maize under drought stress [56]. Like drought stress in maize, when the P138 and EMS-104 seedlings were treated with 150 μM MT application under 100 mM Na_2_CO_3_ stress, their Pn, Gs, and Tr increased significantly, while Ci decreased significantly (Figure 4A–D).

During the evolution of plants, they have developed powerful antioxidant enzyme systems, such as SOD, CAT, POD, and APX, which synergistically enhance plant stress resistance and trigger protective response against oxygen toxicity in cells and throughout the plant [57]. Many studies indicated that MT treatment under abiotic stress boosted the activities of these antioxidant enzymes [58,59,60]. Consistently, our study also showed that the activities of SOD, POD, CAT, and APX in the leaves of P138 and EMS-104 seedlings were significantly increased under 100 mM Na_2_CO_3_ stress and 150 μM MT treatment (Figure 2A–D). At the same time, the expression levels of the four antioxidant enzyme genes, i.e., *Zm00001d025106*, *Zm00001d031908*, *Zm00001d027511*, and *Zm00001d040364,* were up-regulated in P138 and EMS-104 seedlings under MT stimulation and Na_2_CO_3_ stress (Figure 8A–D). Hence, this suggested that 150 μM MT application improved the salt tolerance of maize seedlings by enhancing antioxidant enzyme activities.

In conclusion, under 100 mM Na_2_CO_3_ stress, 150 μM MT application elevated RWC, Pn, Gs, Tr, and Chl content, while reducing the REC, Ci, and H_2_O_2_ content of leaves in maize seedlings. Meanwhile, the six candidate genes related to photosynthetic pigments and antioxidant homeostasis were up-regulated. They then formed complex interaction relationships to enhance salt tolerance in maize seedlings. Briefly, 150 μM MT application maintains the integrity of photosynthetic organs, scavenges excess ROS, and improves photosynthetic characteristics, subsequently alleviating damage and promoting the growth of maize seedlings under a Na_2_CO_3_ stress environment (Figure 9).

## 4. Materials and Methods

### 4.1. Materials

Previously, a total of 1041 M_3_ maize mutant lines were constructed with pollen from the elite inbred line P138, which was treated with 0.5 mg·L^−1^ ethyl methane sulfonate solution (EMS) by our team. P138 is a representative inbred line, derived from the P group, with weak salt tolerance [61]. The mutant EMS-104 is more salt tolerant than the wild-type P138. Therefore, we conducted a follow-up study on P138 and its mutant EMS-104.

### 4.2. Plant Growth Conditions

The seeds of P138 and EMS-104 were sterilized with 0.5% (*v*/*v*) sodium hypochlorite solution for 15 min, rinsed five times with ddH_2_O, and then soaked in ddH_2_O for 24 h at 22 ± 0.5 °C. Subsequently, the ten soaked seeds were planted in sterilized vermiculite plastic boxes (13 cm diameter × 11 cm high), and then cultured in a greenhouse (25 ± 0.5 °C temperature; 300 μM·m^−2^ s^−1^ light intensity; 65% relative humidity; 12/12 h light/dark) for 15 d. During this period, the seedlings were washed by 50 mL ddH_2_O every two days. The three-leaf seedlings received four treatments for 7 d, i.e., 0 mM Na_2_CO_3_ + 0 μM MT (ddH_2_O, CK), 100 mM Na_2_CO_3_ + 0 μM MT(N), 0 mM Na_2_CO_3_ + 150 μM MT(M), and 100 mM Na_2_CO_3_ + 150 μM MT(MN). During all the treatments, the seedlings were washed by 50 mL of the above corresponding mixed solution every two days. Each treatment was repeated three times.

### 4.3. Growth Phenotype Measurements

For the seedlings of the two maize materials under four treatments, their seedling length (SL), seedling weight (SW), and root weight (RW) were measured according to Zhao et al.’s [62] method. The seedling leaf area (LA) was calculated as follows:LA = Leaf length × Leaf width × 1.5.

### 4.4. Stomatal Morphology in Seedlings

The 3rd fresh leaves of P138 and EMS-104 seedlings under all treatments were cut in the middle of the leaf (1 cm × 1 cm), and were stuck on a slide coated with glue for 20 min. The leaf sections were then delicately detached by tweezers, leaving the temporary stomatal slides for collection. Stomatal morphology was examined utilizing an integrated forward and inverted fluorescence microscope (Revolve RVL-100-G, ECHO, San Diego, CA, USA). For each treatment, five random fields of view were selected from the paraxial/dorsal surface. Stomatal length, width, and area were measured and statistically recorded under a 4× objective (image size 3226 × 3024).

### 4.5. Photosynthetic Parameter Determination

The four photosynthetic parameters, including net photosynthetic rate (Pn; μM CO_2_·m^−2^·s^−1^), intercellular CO_2_ (Ci; μM CO_2_·mol^−1^), stomatal conductance (Gs; M H_2_O·m^−2^·s^−1^), and transpiration rate (Tr; mM H_2_O·m^−2^·s^−1^), of the 3rd leaf of P138 and EMS-104 seedlings under all treatments were measured by an LI-6400XT (LI-COR; Biosciences, Inc. Lincoln, NE, USA) portable photosynthetic measurement system. The illuminance inside the leaf chamber was set to 1000 μM/(m^2^·s), a CO_2_ concentration of 400 μM·M^−1^, and a temperature of 25 °C.

### 4.6. Relative Moisture Content Assay

We measured 1.0 g of fresh leaves (FW) from P138 and EMS-104 seedlings under all treatments, which were then completely soaked in 50 mL ddH_2_O until a constant weight (TW) was reached. The soaked leaves were then dried in an oven to a constant weight (DW). The relative water content (RWC) was calculated as follows:[(FW − DW)/(TW − DW)] × 100%.

### 4.7. Relative Electrical Conductivity Measurement

We placed 0.1 g fresh leaves of P138 and EMS-104 seedlings under all treatments in a test tube of 10 mL ddH_2_O, and left them at room temperature for 12 h to determine the electrical conductivity (R_1_). Then, the same set of samples was stored in a 100 °C water bath for 15 min and the electrical conductivity (R_2_) was recorded. The conductivity (R_1_, R_2_) was measured by a DDSJ-308F conductivity meter (Rex Electric Chemical, Shanghai, China). The REC was estimated as follows [62]:REC = (R_1_/R_2_) × 100%.

### 4.8. Chlorophyll Content Measurement

We cut 1.0 g of fresh leaves from P138 and EMS-104 seedlings under all treatments and soaked them in 10 mL of 95% alcohol for 48 h; they were then centrifuged at 12,000 rpm (Centrifuge 5425/5425 R; Eppendorf, Hamburg, Germany) for 10 min. The absorbance of the supernatant was measured by utilizing a multifunctional enzyme marker model (SynergyHTX; BioTek Instruments, Inc., Winooski, VT, USA), with readings taken at wavelengths of 665 nm, 649 nm, and 470 nm. The concentrations of chlorophyll a, chlorophyll b, and carotenoids were calculated by the following formulas: Cac = 13.95 × A665 − 6.88 × A649; Cbc = 24.96 × A649 − 7.32 × A665; and Ccc = 1000 × A470 − 2.05 × Ca – 114 × Cb. The pigment content was calculated by the following equations: Chl a content (mg g^−1^) = Cac × Vt × n/FW × 1000; Chl b content (mg g^−1^) = Cbc × Vt × n/FW × 1000; and Car content (mg/g) = Ccc × Vt × n/FW × 1000. Cac, Cbc, and Ccc represent the concentrations of Chl a, Chl b, and Car, respectively. FW is the fresh weight (g), Vt is the total volume of extract (mL), and n is the dilution factor.

### 4.9. Antioxidant Enzyme Activity Determination

Referring to Zhao et al.’s [62] method, the SOD activity, POD activity, CAT activity, APX activity, and H_2_O_2_ content were determined using corresponding Solarbio kits (Beijing Solarbio Science and Technology Co., Ltd., Beijing, China) and a multifunctional microplate reader (SynergyHTX; BioTek Instruments, Inc., USA), following the manufacturers’ instructions.

### 4.10. qRT-PCR Analysis

The total RNA was extracted from eight samples of the 3rd leaf of salt-sensitive P138 and salt-resistant EMS-104 seedlings (three biological replicates for each sample) under the above four treatments with TRIZOL reagent (Invitrogen, Carlsbad, CA, USA), which was reverse-transcribed into cDNA using a SuperScript III First strand Kit (Invitrogen, Gibco, Grand Island, NE, USA). qRT-PCR was conducted using TransStart Tip Green qPCR SuperMix (Tran, Beijing, China). Primers (Table 1) for six candidate genes involved in antioxidant enzyme activities and photosynthetic pigment biosynthesis were designed via Primer3web v.4.1.0 (https://primer3.ut.ee/; accessed on 13 May 2024). The positions of these genes were mapped in the *Zea mays* B73_V4 reference genome (https://www.maizegdb.org/; accessed on 13 May 2024), and their functional annotations were performed using the tool AgBase v2.00 (https://agbase.arizona.edu/; accessed on 13 May 2024). Relative gene expression levels were calculated by the 2^−∆∆Ct^ method, with *Zm00001d010159* as an internal reference gene.

### 4.11. Statistical Analysis

For all tested traits of the two maize genotypes under all treatments, three-way ANOVAs were performed using IBM-SPSS Statistics v.20.0 software (SPSS, Chicago, IL, USA; https://www.Ibm.com/products/spss-statistics; accessed on June 16 2024). Pearson correlation among all traits was analyzed using Origin 2021 (v. 21.0, OriginPro, Northampton, MA, USA; https://www.genescloud.cn; accessed on 16 June 2024).

## Figures and Tables

**Figure 7 plants-13-02844-f007:**
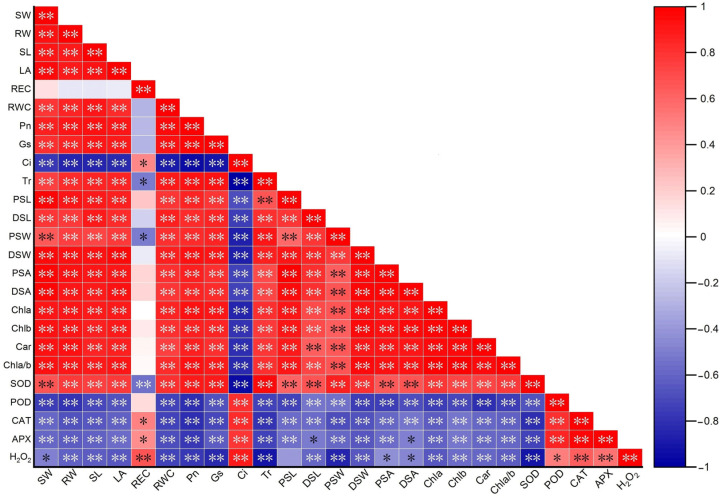
The correlation analysis of 25 tested traits of maize seedlings, including seedling length (SL), leaf area (LA), fresh seedling weight (SW), fresh root weight (RW), relative electrical conductivity (REC), and relative water content (RWC), net photosynthetic rate (Pn), stomatal conductance (Gs), intercellular CO_2_ concentration (Ci), transpiration rate (Tr), paraxial surface stomatal length (PSL), paraxial surface stomatal width (PSW), paraxial surface stomatal area (PSA), dorsal surface stomatal length (DSL), dorsal surface stomatal width (DSW), dorsal surface stomatal area (DSA), chlorophyll a content (Chl a), chlorophyll b content (Chl a/b), carotenoid content (Car), chlorophyll a/b ratio (Chl a/b), superoxide dismutase (SOD), peroxidase (POD), catalase (CAT), ascorbate peroxidase (APX), and H_2_O_2_ content (H_2_O_2_). ** indicates a significant correlation at the *p* < 0.01 level and * indicates a significant association at the *p* < 0.05 level. The red squares represent a positive correlation between the two metrics, and the blue squares represent a negative correlation. The strength of the correlation is indicated by the color of the blocks.

**Figure 8 plants-13-02844-f008:**
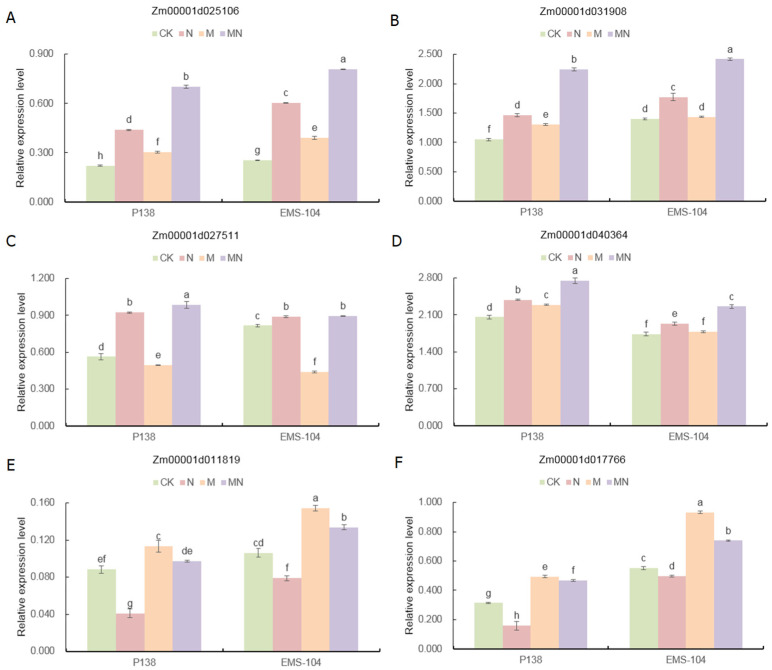
Changes in expression levels of six candidate genes, including superoxide dismutase (*SOD; Zm00001d025106* (**A**), *Zm00001d031908* (**B**), catalase 2 (*CAT2; Zm00001d027511*) (**C**), peroxidase 72 (*POD72; Zm00001d040364*) (**D**), chlorophyllide a oxygenase chloroplastic (*Zm00001d011819*) (**E**), and nine-cis-epoxycarotenoid dioxygenase8 (*Zm00001d017766*) (**F**) in leaves of P138 and EMS-104 seedlings under different treatments: 0 mM Na_2_CO_3_ + 0 μM melatonin (MT) treatment (CK); 100 mM Na_2_CO_3_ + 0 μM MT treatment (N); 0 mM Na_2_CO_3_ + 150 μM MT treatment (M); and 100 mM Na_2_CO_3_ + 150 μM MT treatment (MN). Different lowercase letters indicate significant differences in *p* < 0.05 level.

**Figure 9 plants-13-02844-f009:**
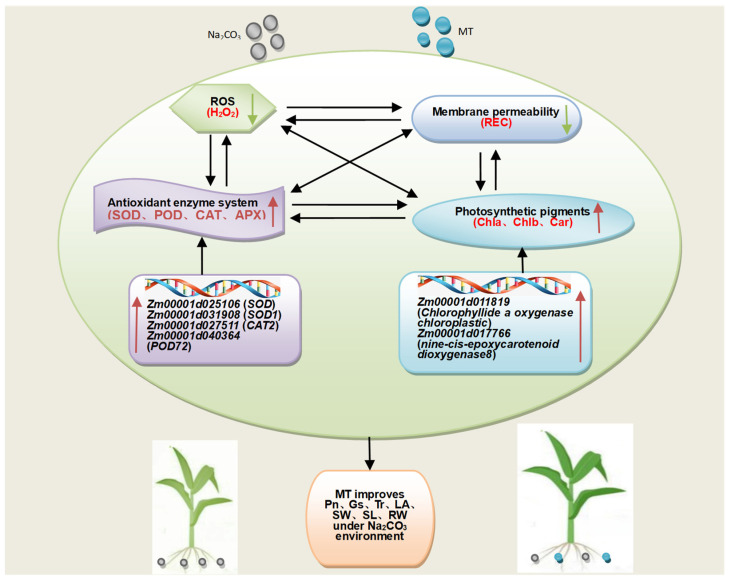
The response mechanism of melatonin (MT) application in maize seedlings under Na_2_CO_3_ stress. Chlorophyll a content (Chl a); chlorophyll b content (Chl b); chlorophyll a:b ratio (Chl a/b); photosynthetic performance, including net photosynthetic rate (Pn), stomatal conductance (Gs), and transpiration rate (Tr); antioxidant enzyme system, namely superoxide dismutase activity (SOD), peroxidase activity (POD), catalase activity (CAT), and ascorbate peroxidase activity (APX); reactive oxygen species (ROS) levels; H_2_O_2_ content (H_2_O_2_); membrane permeability (MP); relative electrical conductivity (REC). The red and green arrows indicate increases and decreases in the corresponding traits, respectively.

**Table 1 plants-13-02844-t001:** Sequences of primers used in reverse-transcription quantitative PCR (qRT-PCR) and functional annotation of six candidate genes.

Gene ID (Encoded Protein)	Gene Position	Primer Sequence (5′ to 3′)	Gene Functional Annotation
*Zm00001d025106* (*superoxide dismutase*)	Chromosome 10 (104541524_104545523 bp)	F: TTGAACTTCACTGGGGTAAGC R: ACAAAAGACTCTGCACGCATC	Superoxide dismutase activity (GO:0004784); removal of superoxide radicals (GO:0019430); superoxide metabolic process (GO:0006801)
*Zm00001d031908* (*superoxide dismutase (SOD1a)*)	Chromosome 1 (206405747_206408926 bp)	F: TTCGCCGCTCCCTATTCC R: GTCCTGTCGATATGCACCCA	Superoxide dismutase activity (GO:0004784); oxidoreductase activity (GO:0016491); antioxidant activity (GO:0016209)
*Zm00001d027511* (*catalase 2*)	Chromosome 1 (7144389_7146665 bp)	F: CCCCAACTACCTGCTGCTAC R: TGGTTATGAACCGCTCTTGC	Oxidoreductase activity (GO:0016491); catalase activity (GO:0004096); peroxidase activity (GO:0004601); response to abiotic stimulus (GO:0009628); response to hormone (GO:0009725); response to reactive oxygen species (GO:0000302); cellular oxidant detoxification (GO:00098869); hydrogen peroxide catabolic process (GO:0042744); response to oxidative stress (GO:0006979); response to hydrogen peroxide (GO:0042542); peroxisome (GO:0005777)
*Zm00001d040364* (*peroxidase 72*)	Chromosome 3 (40090724_40092823 bp)	F: GGATGTATCCTACGCCGCAA R: TTGTCAAACTTGGCAGGGGT	Oxidoreductase activity (GO:0016491); peroxidase activity (GO:0004601); cellular oxidant detoxification (GO:0098869); response to oxidative stress (GO:0006979); hydrogen peroxide catabolic process (GO:0042744)
*Zm00001d011819* (*chlorophyllide a oxygenase chloroplastic*)	Chromosome 8 (162062135_162065501)	F: CCATCAAGAAGGGCAAGTTCC R: TCTTTCCTCAAGGTCCCGAT	Oxidoreductase activity (GO:0016491); chlorophyllide a oxygenase [overall] activity (GO:0010277); chlorophyll biosynthetic process (GO:0015995); plastid (GO:0009536); chloroplast (GO:0009507); cytoplasm (GO:0005737)
*Zm00001d017766* (*nine–cis–epoxycarotenoid dioxygenase8*)	Chromosome 5 (206198899_206201211 bp)	F: CCACGCACACCAGAGTTACA R: GCTGGGCGCCTTTCTACTAA	Carotene catabolic process (GO:0016121); Obsolete oxidation–reduction process (GO:0055114); Chloroplast (GO:0009507); Chloroplast stroma (GO:0009570); Carotenoid dioxygenase activity (GO:0010436); oxidoreductase activity, acting on single donors with incorporation of molecular oxygen, incorporation of two atoms of oxygen (GO:0016702); 9–cis–epoxycarotenoid dioxygenase activity (GO:0045549); metal ion binding (GO:0046872); dioxygenase activity (GO:0051213)
*Zm00001d010159* (*actin 1*)	Chromosome 8 (102413768_102417536 bp)	F: CGATTGAGCATGGCATTGTCA R: CCCACTAGCGTACAACGAA	ATP binding (GO:0005524); nucleotide binding (GO:0000166)

## Data Availability

Data are contained within the article and Appendix A.

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
