# Peer review of "Exogenous Melatonin Reinforces Photosynthesis, Antioxidant Defense and Gene Expression to Ameliorate Na2CO3 Stress in Maize"

_plants, 2024, doi:10.3390/plants13202844_

Round 1

Reviewer 1 Report

Comments and Suggestions for Authors

The present study explores the protective effect of melatonin on various parameters of maize plants in response to Na2CO3 stress. The paper will be interesting for readers of Plants journal but before being suitable for publication the deficiencies in the MS must be resolved.

1. A big issue is a language that needs to be improved. Sometimes it’s difficult to understand what the authors wanted to write.

For instance: P. 1-2. L. 44-45.

“Whereas its susceptibility to salt stress, particularly during the seedling stage [3]. İsmail et al. [4] reported that maize is the most salt-sensitive among cereal crops, with its maximum salinity tolerance at 0.017 M.”

The sentences are grammatically incorrect. Then, I didn’t find information about this concentration (and it’s unclear what does this refer to) in report by İsmail et al.

There are many other places where the language must be corrected. I encourage the authors to double check the entire text with the help of a native speaker or professional proofreader to improve the language.

2. Fig. 4.

I think the inscriptions on the Y axis should be unified – the authors used Ci abbreviation in Fig. 4C. Why they don’t use abbreviations in other subfigures?

3. P. 3-4, L. 135-137.

“…large amounts of H2O2 induced by Na2CO3 stress broke the homeostasis of reactive oxygen species, and then caused oxidative stress.???

Maybe “…Na2CO3 stress broke the homeostasis of reactive oxygen species leading to accumulation of large amounts of H2O2 thus causing the oxidative stress.”

4. P. 4, L. 137-138.

“The antioxidant enzymes in the cells of maize seedlings would be produced in large quantities to remove excess H2O2. Under M treatment, SOD content of P138 increased by 24.9%, APX decreased by 19.2%, POD and CAT content had no significant difference.”

SOD don’t scavenge H2O2. Here, it is more appropriate to write ROS.

5. P. 9, L. 317-319.

Why did you discuss plant growth-promoting bacteria?

6. P. 10, L. 327-331.

The claims made here are questionable.

“The increase of photosynthetic pigments helps to enhance the resistance to ROS…”?

How is this possible? Anyway, there is no mention of photosynthetic pigments in the reference by Li et al. (28).

6. P. 10, L. 343-348.

What is described here is a repetition of what was in the results. And it's boring to read about percentage changes again.

7. P. 10, L. 351-354.

Again, the claims made here are very speculative. What are these statements based on? And why not to discuss the results obtained by the authors themselves on stomata?

8. P. 10, L. 355-364.

This may be shortened.

9. Overall, the Discussion section is not consistent and seems weak. I highly recommend rewriting it. This section strangely begins with a discussion of photosynthetic pigments. I think the Discussion should start with the problem that the authors have raised. There is no need to list percentage changes here. But the authors must refer to Figures. Some results are not discussed or are discussed weakly (sections 2.1, 2.3, 2.6, and 2.7). There are no thoughts on differences between varieties.

10. P. 13, L. 482.

…“treatmentserials” …?

11. The Materials and Methods

Several subsections should be rewritten or considerably revised. It is necessary to correct the writing style of the MM section. This is not the manual. The authors need to correct the subsections written in the imperative mood. Then, the authors should pay attention to the language.

12. The Materials and Methods

Why these concentrations of Na2CO3 and MT were used?

Comments on the Quality of English Language

Sometimes it’s difficult to understand what the authors wanted to write. I encourage the authors to double check the entire text with the help of a native speaker or professional proofreader to improve the language.

Author Response

Thank you for your letter of – and for the referee’s comments concerning our manuscript, “Exogenous Melatonin Reinforces Photosynthesis, Antioxidant Defense, and Their Genes Expression to Ameliorate Na2CO3 Stress in Maize (Manuscript ID: plants-3194664)”. We have carefully studied these comments and have made corresponding corrections to the manuscript, which we describe in detail below. We would like to re-submit the manuscript and that for possible publication on the Special Issue: “Effect of Biotic and Abiotic Factors on the Physiology of Horticultural Plants” of Plants. Thank you very much for your time and consideration.

Editor:

Your manuscript has now been reviewed by experts in the field and can be found with the review reports at: https://susy.mdpi.com/user/manuscripts/resubmit/fa18f33c95d35dc6f3ee60b75add3ffe Please revise the manuscript found at the above link according to the reviewers' comments and upload the revised file within 10 days.

Thanks for the positive comments of you and all reviewers for our manuscript. As suggested, we have carefully revised and improved our manuscript using the “Track Changes” function of the manuscript at the above link. We then have re-submitted the manuscript within the allotted time.

Thank you for your consideration.

(I) Ensure all references are relevant to the content of the manuscript.

Thanks for the positive comments. As suggested, we have carefully checked all references. We then have re-submitted the manuscript.

Thank you for your consideration.

(II) Highlight any revisions to the manuscript, so editors and reviewers can see any changes made.

Thanks for the positive comments. As suggested, we have carefully revised and improved our manuscript using the “Track Changes” function of the manuscript. We then have re-submitted the manuscript.

Thank you for your consideration.

(III) Provide a cover letter to respond to the reviewers’ comments and explain, point by point, the details of the manuscript revisions.

Thanks for your positive comments for our manuscript. As suggested, we have carefully revised and improved our manuscript. In addition, we have prepared a detailed response letter to all reviewers for each point, and then have re-submitted the manuscript.

Thank you for your consideration.

(IV) If the reviewer(s) recommended references, critically analyze them to ensure that their inclusion would enhance your manuscript. If you believe these references are unnecessary, you should not include them.

Thanks for your positive comments for our manuscript. As suggested, we have carefully checked and revised the References. At the same time, we also have re-added twenty-one new references to enhance the quality of our manuscript. We then have re-submitted the manuscript.

Thank you for your consideration.

 (V) If you found it impossible to address certain comments in the review reports, include an explanation in your appeal.

Thanks for your positive comments for our manuscript. As suggested, we have carefully revised and improved our manuscript. In addition, we have prepared a detailed response letter to all reviewers for each point, and then have re-submitted the manuscript.

Thank you for your consideration.

If your manuscript requires improvement to the language and/or figures, you may consider MDPI Author Services: https://www.mdpi.com/authors/english. Please note the status of this invitation “Publish Author Biography on the webpage of the paper” - https://susy.mdpi.com/user/manuscript/author_biography/b0aa56b7bef0a99dd4fe8c1bf66a47d6. If you wish to publish your biography, please complete it before your manuscript is accepted.

Thanks for the positive comments. As suggested, we have carefully checked and revised the English language of the manuscript. We then re-submitted the manuscript.

In addition, thanks for your invitation, we decided not to publish our biography.

Thank you for your consideration.

Please do not hesitate to contact us if you have any questions regarding the revision of your manuscript or if you need more time. We look forward to hearing from you soon.

Thanks for your positive comments for our manuscript. As suggested, we have carefully revised and improved the manuscript using the “Track Changes” function of our manuscript at the above link. We then have re-submitted the manuscript within the allotted time.

Thank you for your consideration.

Reviewer 1

Comments and Suggestions for Authors

The present study explores the protective effect of melatonin on various parameters of maize plants in response to Na2CO3 stress. The paper will be interesting for readers of Plants journal but before being suitable for publication the deficiencies in the MS must be resolved.

Thanks for your positive comments. As suggested, we have resolves defects in MS. We then re-submitted the manuscript.

Thank you for your consideration.

  1. A big issue is a language that needs to be improved. Sometimes it’s difficult to understand what the authors wanted to write. For instance: P. 1-2. L. 44-45.

Thanks for your positive comments.  As suggested, we reorganized the language.  We then have re-submitted the manuscript.

Thank you for your consideration.

  1. “Whereas its susceptibility to salt stress, particularly during the seedling stage [3]. İsmail et al. [4] reported that maize is the most salt-sensitive among cereal crops, with its maximum salinity tolerance at 0.017 M.” The sentences are grammatically incorrect. Then, I didn’t find information about this concentration (and it’s unclear what does this refer to) in report by İsmail et al.

Thanks for your positive comments. As suggested, we replaced the correct references were that “Generally, maize is extremely sensitive to salt stress during its whole growth period, especially the seedling stage [8,9]” in Lines 44-45 of the manuscript. We then have re-submitted the manuscript.

Thank you for your consideration.

  1. There are many other places where the language must be corrected. I encourage the authors to double check the entire text with the help of a native speaker or professional proofreader to improve the language.

Thanks for your positive comments. As suggested, we have re-edited the whole article in English. We then have re-submitted the manuscript.

Thank you for your consideration.

  1. Fig. 4. I think the inscriptions on the Y axis should be unified – the authors used Ci abbreviation in Fig. 4C. Why they don’t use abbreviations in other subfigures?

Thanks for your positive comments. As suggested, we changed the legend names for the four metrics in Figure 4. We have revised and improved the manuscript. We then have re-submitted the manuscript.

Thank you for your consideration.

Figure 4. Changes of net photosynthetic rate (A), stomatal conductance (B), intercellular CO2 concentration (C), and transpiration rate (D) of P138 and EMS-104 seedlings under different treatments, including 0 mM Na2CO3+0 μM melatonin (MT) treatment (CK); 100 mM Na2CO3+0 μM MT treatment (N); 0 mM Na2CO3+150 μM MT treatment (M); 100 mM Na2CO3+150 μM MT treatment (MN). Different lowercase letters indicated significant differences in p < 0.05 level.

  1. P. 3-4, L. 135-137. “…large amounts of H2O2 induced by Na2CO3 stress broke the homeostasis of reactive oxygen species, and then caused oxidative stress.??? Maybe “…Na2CO3 stress broke the homeostasis of reactive oxygen species leading to accumulation of large amounts of H2O2 thus causing the oxidative stress.”

Thanks for your positive comments. As suggested, we have revised the corresponding contents were that “The results showed that Na2CO3 stress enhances antioxidant enzymes activities to scavenge excess ROS in maize.” in Lines 126-127 of the manuscript. We then have re-submitted the manuscript.

Thank you for your consideration.

  1. P. 4, L. 137-138. “The antioxidant enzymes in the cells of maize seedlings would be produced in large quantities to remove excess H2O2. Under M treatment, SOD content of P138 increased by 24.9%, APX decreased by 19.2%, POD and CAT content had no significant difference.” SOD don’t scavenge H2O2. Here, it is more appropriate to write ROS.

Thanks for your positive comments. As suggested, we have revised the corresponding contents were that “The results showed that Na2CO3 stress enhances antioxidant enzymes activities to scavenge excess ROS in maize. Moreover, the SOD activity of P138 seedlings significantly increased by 24.9%, while the APX activity clearly decreased by 19.2% under M treatment; similarly, the SOD activity increased significantly by 55.8%, however, the CAT activity decreased clearly by 84.9% in EMS-104 seedlings under M treatment (Figure 2A; 2D).” in Lines 126-132 of the manuscript. We then have re-submitted the manuscript.

Thank you for your consideration.

  1. P. 9, L. 317-319. Why did you discuss plant growth-promoting bacteria?

Thanks for your positive comments. As suggested, we have revised the corresponding references “Under drought stress, Ahmad et al. [44] discovered that chlorophyll levels were clearly increased in maize after 100 μM MT application. For other exogenous hormones application, similar results were confirmed that exogenous salicylic acid (SA) increased the chlorophyll content of maize under salt stress [45].” in Lines 329-332 of the manuscript. We then have re-submitted the manuscript.

Thank you for your consideration.

  1. P. 10, L. 327-331. The claims made here are questionable. “The increase of photosynthetic pigments helps to enhance the resistance to ROS…”? How is this possible? Anyway, there is no mention of photosynthetic pigments in the reference by Li et al. (28).

Thanks for your positive comments. As suggested, we have revised the corresponding contents were that “In addition, some studies have shown that there is a positive feedback relationship between the content of chlorophyll and ROS metabolism, the increase of chlorophyll levels helped to scavenge ROS accumulation in plant [46].” in Lines 341-343 of the manuscript. We then have re-submitted the manuscript.

Thank you for your consideration.

  1. P. 10, L. 343-348. What is described here is a repetition of what was in the results. And it's boring to read about percentage changes again.

Thanks for your positive comments. As suggested, we streamlined the content and removed the percentages were that “Our study showed that 100 mM Na2CO3 stress significantly decreased the contents of Chl a, Chl b and Car of leaves in P138 and EMS-104 seedlings. In contrast, 150 μM MT treatment induced significant increases in these photosynthetic pigments (Figure 5A5D)” in Lines 333-335 of the manuscript. We then have re-submitted the manuscript.

Thank you for your consideration.

  1. P. 10, L. 351-354. Again, the claims made here are very speculative. What are these statements based on?

Thanks for your positive comments. As suggested, we decided to delete this part of the article in order to make the article more reasonable. We then have re-submitted the manuscript.

Thank you for your consideration.

  1. And why not to discuss the results obtained by the authors themselves on stomata?

In addition, as suggested, we have supplemented our experimental study on stomata were that “The changes of stomatal morphology are positive physiological responses on different environmental conditions [47]. Some studies have shown that salt stress led to stomatal closure, and exogenous Si treatment improved the Gs of alfalfa (Medicago sativa L.) seedlings under salt stress [48]. Similarly, exogenous SA application improved the Gs of barley under salt stress [49]. We also found that the application of 150 μM MT improved the paraxial/dorsal surface stomatal morphology under 100 mM Na2CO3 stress in current study (Figure 6A6F).” in Lines 349-355 of the manuscript. We then have re-submitted the manuscript.

Thank you for your consideration.

  1. P. 10, L. 355-364. This may be shortened.

Thanks for your positive comments. As suggested, we have revised the corresponding contents were that “During the evolution of plants, they have developed powerful antioxidant enzyme systems, such as SOD, CAT, POD, and APX, which synergistically enhanced plant stress resistance and trigger protective responses against oxygen toxicity in cells and throughout the plant [58]. Many researches indicated that MT treatment under abiotic stress boosted the activities of these antioxidant enzymes [5961].” in Lines 367-371 of the manuscript. We then have re-submitted the manuscript.

Thank you for your consideration.

  1. Overall, the Discussion section is not consistent and seems weak. I highly recommend rewriting it. This section strangely begins with a discussion of photosynthetic pigments. I think the Discussion should start with the problem that the authors have raised. There is no need to list percentage changes here. But the authors must refer to Figures. Some results are not discussed or are discussed weakly (sections 2.1, 2.3, 2.6, and 2.7).

Thanks for your positive comments. As suggested, we have rewritten the discussion, enter from the questions we studied, added the parts that were not discussed or discussed very weakly, removed the percentage changes, and referenced the charts were that “It is well known that salt stress is the major environmental factor in the soil that inhibits and negatively affects plant growth, development, and yield formation [30,31]. Fortunately, different strategies have been developed in some plant species to retort various environmental stresses. Previous studies have shown that exogenous MT application could enhance salt tolerance in rice, soybean, cucumber, and sunflower [21–24]. However, our knowledge regarding the mechanisms involved in MT-mediated tolerance to Na2CO3 stress of salt-susceptible P138 and its salt-resistant EMS-104 mutant seedlings still remains unclear.

Phenotypes were directly observed that salt stress negatively affected the biomass and plant height of plants [32–34]. The 150 μM MT increased the leaf area and biomass of cotton (Gossypium hirsutum L.) plants under salt stress [35]. A similar phenomenon was also found in our study, i.e., under the stress of 100 mM Na2CO3, the seedling length, leaf area, seedling weight, and root weight of P138 and EMS-104 mutant were significantly reduced (Figure 1A1D). Indicating that salt stress severely inhibits the survival or health growth of maize. while we also found that the growth status of EMS-104 mutant was better than its wild P138 (Figure 1A1D), which provides valuable genetic resources for salt tolerance breeding in future. After applying 150 μM MT, these growth phenotypes were improved significantly (Figure 1A1D). It is thus speculated that MT application may improve multiple physiological processes of maize seedlings to resist Na2CO3 stress injury.

The RWC can directly reflect water status of plants [36], and the REC is generally considered to show membrane permeability and membrane integrity under stress [37]. Zhao et al. [38] found that drought stress reduced RWC of maize, and Wang et al. [39] found that salt stress combined with drought stress increased REC of boxwood seedlings. Interestingly, early studies have shown that the application of MT was beneficial for maintaining intracellular turgor pressure and improving plasma ion homeostasis [39,40]. In this study, we found that RWC was significantly decreased in both maize genotypes of P138 and EMS-104 leaves, while their REC was significantly increased under the stress of 100 mM Na2C03, which was consistent with the results of previous studies [38,39]. After applying 150 μM MT, their RWC of P138 and EMS-104 seedlings were significantly increased, but their REC were significantly decreased (Figure 3A3B). Indicating that 150 μM MT treatment may maintain relatively high osmotic pressure to protect cell membrane integrity in maize seedlings under 100 mM Na2C03 stress.

Photosynthetic pigments are closely related to photosynthesis and stress resistance of plants [41,42]. Salt stress accelerated the degradation of chlorophyll and inhibited the biosynthesis of chlorophyll in maize [43]. Under drought stress, Ahmad et al. [44] discovered that chlorophyll levels were clearly increased in maize after 100 μM MT application. For other exogenous hormones application, similar results were confirmed that exogenous salicylic acid (SA) increased the chlorophyll content of maize under salt stress [45]. Our study showed that 100 mM Na2CO3 stress significantly decreased the contents of Chl a, Chl b, and Car of leaves in P138 and EMS-104 seedlings. In contrast, 150 μM MT treatment induced significant increases in these photosynthetic pigments (Figure 5A5D). This is consistent with the results of previous studies [44,45]. In addition, our gene expression analysis also showed that the Zm00001d017766 and Zm00001d011819, two photosynthetic pigments related genes had positive expression level under MT application in both maize genotypes (Figure 8E8F). Therefore, these findings indicated that 150 μM MT could promote photosynthetic pigments genes expression to increase the photosynthetic pigments accumulation under 100 mM Na2CO3 stress. In addition, some studies have shown that there is a positive feedback relationship between the content of chlorophyll and ROS metabolism, the increase of chlorophyll levels helped to scavenge ROS accumulation in plant [46]. We also found that ROS levels, such as H2O2 content were significantly decreased in P138 and EMS-104 seedlings when they treated with 150 μM MT application under 100 mM Na2CO3 stress (Figure 2E). Thereby, the lower ROS level may benefit for photosynthetic pigments production and maintain high photosynthesis in maize under various environmental stimulations.

The changes of stomatal morphology are positive physiological responses on different environmental conditions [47]. Some studies have shown that salt stress led to stomatal closure, and exogenous Si treatment improved the Gs of alfalfa (Medicago sativa L.) seedlings under salt stress [48]. Similarly, exogenous SA application improved the Gs of barley under salt stress [49]. We also found that the application of 150 μM MT improved the paraxial/dorsal surface stomatal morphology under 100 mM Na2CO3 stress in current study (Figure 6A6F).

Photosynthesis is the fundament for a plants' survival, growth, reproduction, and resilience to environmental stresses [50,51]. Salt stress reduced light energy absorption and conversion efficiency of leaves [52]. Previous studies showed that exogenous MT protected the structure chloroplasts and photosynthetic organs, induced stomatal reopening, sustained high levels of chlorophyll, and enhanced the leaf ability to capture light energy, this action thus alleviated the inhibition of photosynthesis caused by NaCl stress in alfalfa leaves [53–56]. Similar studies also found that the MT stimulation significantly enhanced the photosynthetic efficiency in maize under drought stress [57]. Like drought stress in maize, when the P138 and EMS-104 seedlings were treated with 150 μM MT application under 100 mM Na2CO3 stress, their Pn, Gs, and Tr were increased significantly, while Ci was decreased significantly (Figure 4A4D).

During the evolution of plants, they have developed powerful antioxidant enzyme systems, such as SOD, CAT, POD, and APX, which synergistically enhanced plant stress resistance and trigger protective responses against oxygen toxicity in cells and throughout the plant [58]. Many researches indicated that MT treatment under abiotic stress boosted the activities of these antioxidant enzymes [5961]. Consistently, our study also showed that the activities of SOD, POD, CAT, and APX in the leaves of P138 and EMS-104 seedlings were significantly increased under 100 mM Na2CO3 stress and 150 μM MT treatment (Figure 2A2D). At the same time, the expression levels of the four antioxidant enzymes genes, i.e., Zm00001d025106, Zm00001d031908, Zm00001d027511, and Zm00001d040364 were up-regulated in P138 and EMS-104 seedlings under MT stimulation and Na2CO3 stress (Figure 8A8D). Hence, which suggested that 150 μM MT application improves the salt tolerance of maize seedlings by enhancing the antioxidant enzymes activities.

In conclusion, under 100 mM Na2CO3 stress, 150 μM MT application elevated RWC, Pn, Gs, Tr, and Chl content, while reduced REC, Ci, and H2O2 content of leaves in maize seedlings. Meanwhile, the six candidate genes related to photosynthetic pigments and antioxidant homeostasis were up-regulated. They then formed complex interaction relationships to enhance salt tolerance in maize seedlings. Briefly, 150 μM MT application maintains the integrity of photosynthetic organs, scavenging excess ROS, and improving photosynthetic characteristics, subsequently alleviating damage and promoting the growth of maize seedlings under Na2CO3 environment (Figure 9).” in Lines 295-386 of the manuscript. We then have added the relevant content re-submitted the manuscript.

Thank you for your consideration.

  1. There are no thoughts on differences between varieties.

Thanks for your positive comments. As suggested, we have revised the corresponding contents were that “while we also found that the growth status of EMS-104 mutant was better than its wild P138 (Figure 1A1D)” in Lines 309-310 of the manuscript. We then have added the relevant content and re-submitted the manuscript.

Thank you for your consideration.

  1. P. 13, L. 482. …“treatmentserials” …?

Thanks for your positive comments. As suggested, we have revised the corresponding contents were that “treatments” in Lines 469 of the manuscript. We then have re-submitted the manuscript.

Thank you for your consideration.

The Materials and Methods

16.Several subsections should be rewritten or considerably revised. It is necessary to correct the writing style of the MM section. This is not the manual. The authors need to correct the subsections written in the imperative mood.

Thanks for your positive comments. As suggested, we have revised the MM section contents were that “4.1. Materials

Previous 1,041 M3 maize mutant lines were constructed by elite inbred line P138 pollen, which was treated with 0.5 mg · L-1 ethyl methane sulfonate solution (EMS). P138 is a representative inbred line, derives from the P group, with weak salt tolerant [62]. The mutant EMS-104 was more salt tolerant than the wild-type P138. The P138 and its mutant EMS-104 thus were performed in following study.

4.2. Plant Growth Conditions.

The seeds of P138 and EMS-104 were sterilized with 0.5% (v/v) sodium hypochlorite solution for 15 min, rinsed five times with ddH2O, and then soaked in ddH2O for 24 h at 22±0.5°C environment. Subsequently the ten soaked seeds were planted in sterilized vermiculite plastic boxes (13 cm diameter × 11 cm high), they then cultured in a greenhouse (25±0.5°C temperature; 300 μM · m-2 s-1 light intensity; 65% relative humidity; 12/12 h light/dark) for 15 d, During this period, the seedlings was washed 50 mL ddH2O every two days. The three-leaf seedlings were then treated four treatments for 7 d, i.e., 0 mM Na2CO3+0 μM MT (ddH2O, CK), 100 mM Na2CO3+0 μM MT(N), 0 mM Na2CO3+150 μM MT(M), and 100 mM Na2CO3+150 μM MT(MN). During the all treatments, the seedlings was washed 50 mL above corresponding mixed solution every two days. Each treatment was repeated three times.

4.3. Growth Phenotypes Measurements

For the seedlings of the two maize materails under four treatments, their seedling length (SL), seedling weight (SW), and root weight (RW) were measured according to Zhao et al [63] method. The seedling leaf area (LA) was calculated as follows:

LA = Leaf length × Leaf width × 1.5.

4.4. Stomatal Morphology in Seedlings

The 3rd fresh leaf of P138 and EMS-104 seedlings under all treatments was cut middle of leaf (1cm × 1 cm), Then a slide coated with glue is attached. Wait for 20 minutes and then use tweezers to gently remove the leaf and collect temporary slides of stomata. Stomatal morphology was observed by a Forward and inverted integrated fluorescence microscope (Revolve RVL-100-G,ECHO USA). Five fields of view were randomly selected for each treatment on the front and back sides. Stomatal length, width, and area were measured and statistically recorded under a 4x objective (image size 3226 × 3024).

4.5. Photosynthetic Parameters Determination

The four photosynthetic parameters, including net photosynthetic rate (Pn; μM CO2·m-2·s-1), intercellular CO2 (Ci; μM CO2·mol-1), stomatal conductance (Gs; M H2O·m-2·s-1), and transpiration rate (Tr; mM H2O·m-2·s-1) of the 3rd leaf in P138 and EMS-104 seedlings under all treatments was measured using a LI-6400XT (LI-CORIn, USA) portable photosynthetic measurement system. Set the illuminance inside the leaf chamber to 1000 μM/ (m2 · s), CO2 concentration of 400 μM · M-1, at the temperature of 25°C.

4.6. Relative Moisture Content Assay

The 1.0 g fresh leaves (FW) of P138 and EMS-104 seedlings under all treatments was measured, and which then were completely soak in 50 mL ddH2O until constant weight (TW). The soaked leaves are then dried in an oven to constant weight (DW). The relative water content (RWC) was calculated as follows:

[(FW-DW)/(TW-DW)] ×100%.

4.7. Relative Electrical Conductivity Measurement

The 0.1 g fresh leaves of P138 and EMS-104 seedlings under all treatments were placed in a test tube of 10 mL ddH2O and placed at room temperature for 12 h to determine the electrical conductivity (R1). Then, the same set of samples were stored in a 100°C water bath for 15 min and the electrical conductivity (R2) was recorded. The conductivity (R1, R2) was measured by DDSJ-308F conductivity meter (Rex Electric Chemical, Shanghai, China). The REC was estimated as follows [63]:

REC = (R1 / R2) × 100%.

4.8. Chlorophyll Content Measurement

The 1.0 g of fresh leaves of P138 and EMS-104 seedlings under all treatments, soaking them in 10 mL of 95% alcohol for 48 h, then the absorbance value of leach liquor was measured using a multifunctional enzyme marker model (SynergyHTX, USA) at 665 nm, 649 nm, and 470 nm. The concentrations of chlorophyll a, chlorophyll b, and carotenoids were calculated using the following formulas: Cac = 13.95×A665 - 6.88×A649, Cbc = 24.96×A649 - 7.32×A665, and Ccc = 1000×A470 - 2.05×Ca - 114×Cb. The pigment content was calculated using the following equations: Chl a content (mg g-1) = Cac × Vt × n / FW × 1000, Chl b content (mg g-1) = Cbc × Vt × n / FW × 1000, and Car content (mg/g) = Ccc × Vt × n / FW × 1000. Where, the Cac, Cbc, and Ccc represented the concentrations of Chl a, Chl b, and Car, respectively. FW was the fresh weight (g), and Vt was the total volume of extract (mL), and n was the dilution factor.

4.9. Antioxidant Enzyme Activity Determination

Refer to Zhao et al. [63] method, the SOD activity, POD activity, CAT activity, APX activity, and H2O2 content were determined using corresponding Solarbio kits (Beijing Solarbio Science and Technology Co., Ltd., Beijing, China) and using the multi-function microplate reader (SynergyHTX; BioTek Instruments, Inc. USA), following the manufacturer’s kit instructions, respectively.

4.10. qRT-PCR Analysis

The total RNA was extracted from eight samples of the 3rd leaf of salt-sensitive P138 and salt-resistant EMS-104 seedlings (three biological replicates for each sample) under above four treatments with TRIZOL reagent (Invitrogen, USA), which was reverse-transcribed into cDNA using a SuperScript III First strand Kit (Invitrogen, Gibco). The qRT-PCR was conducted using TransStart Tip Green qPCR SuperMix (Tran, Beijing, China). Primers (Table 1) for six candidate genes involved in antioxidant enzymes activities and photosynthetic pigments biosynthesis were designed via Primer3web v.4.1.0 (https://primer3.ut.ee/; accessed on 13 May 2024). The positions of these genes were mapped in the Zea mays B73_V4 reference genome (https://www.maizegdb.org/; accessed on 13 May 2024), and their functional annotations were performed using the tool AgBase v2.00 (https://agbase.arizona.edu/; accessed on 13 May 2024). Relative gene expression level was calculated by the 2−∆∆Ct method, with Zm00001d010159 as an internal reference gene.

4.11. Statistical Analysis

For all tested traits of the two maize genotypes under all treatments, their one-way ANOVA was performed using the BM-SPSS Statistics v.20.0 software (SPSS, Chicago, IL., USA; https://www.Ibm.com/products/spss-statistics; accessed on June 16 2024). Pearson correlation among all traits were analyzed the Origin 2021(v. 21.0, OriginPro, USA; https://www.genescloud.cn; accessed on June 16 2024).” in Lines 397-486 of the manuscript. We then have re-submitted the manuscript.

Thank you for your consideration.

  1. The Materials and Methods

Why these concentrations of Na2CO3 and MT were used?

Thanks for your positive comments. As suggested, we quite agree with you. According to previous work in our team (Bo, W.W.; Cao, L.; Chen, W.N.; Lu, C.F.; Hu, Z.H.; Leng, P.H. Quercus dentata responds to Na2CO3 stress with salt crystal deposits: ultrastructure, and physiological–biochemical parameters of leaves. Trees 2023, 4. 10011011.), they screened out the suitable concentration of Na2CO3 stress in maize was 100 mM Na2CO3. Therefore, in this study, we used the 100 mM Na2CO3 concentration to perform our study.

Before we start our this experiment, we read a large of reference in recent years, we found that the optimum concentration of exogenous melatonin (MT) was 100~180 μM, that can significant improve various stresses injury (drought, reduces Cd absorption) in different plants (Ahmad, S., Wang, G.Y., Muhammad, I., Farooq, S., Kamran, M., Ahmad, I., Zeeshan, M., Javed, T., Ullah, S., Huang, J.H. and Zhou, X.B., Application of melatonin-mediated modulation of drought tolerance by regulating photosynthetic efficiency, chloroplast ultrastructure, and endogenous hormones in maize. Chemical and Biological Technologies in Agriculture, 2022, 9, 1-14; Xu, L., Xue, X., Yan, Y., Zhao, X., Li, L., Sheng, K. and Zhang, Z. Silicon Combined with Melatonin Reduces Cd Absorption and Translocation in Maize. Plants, 2023,12, 3537.), especially, wang (Wang H. Response of melatonin at different concentrations to loquat seedlings under cold stress. 2020, 5, 102.) reported that 150 μM MT treatment alleviated the damage of loquat seedlings under low temperature stress; Ahmad et al. (Ahmad, S., Guo YW., Ihsan M., Saqib F., Muhammad K., Irshad A., Muhammad Z.; et al. Application of melatonin-mediated modulation of drought tolerance by regulating photosynthetic efficiency, chloroplast ultrastructure, and endogenous hormones in maize. Chemical and Biological Technologies in Agriculture, 2022, 9, 1-14) showed that 150 μM MT application can improved photosynthetic efficiency, chloroplast ultrastructure, and endogenous hormones levels in maize under drought stress. In these regards, in this study, we selected the 150 μM MT concentration to study the alleviation effects in two maize genotypes seedlings under 100 mM Na2CO3 stress.

Thank you for your consideration.

  1. Then, the authors should pay attention to the language.

Thanks for your positive comments. As suggested, we have re-edited the whole article in English. We then have re-submitted the manuscript.

Thank you for your consideration.

Comments on the Quality of English Language

Sometimes it’s difficult to understand what the authors wanted to write. I encourage the authors to double check the entire text with the help of a native speaker or professional proofreader to improve the language.

Thanks for your positive comments. As suggested, we have re-edited the whole article in English. We then have re-submitted the manuscript.

Thank you for your consideration.

Open Review: I would not like to sign my review report.

Thanks for your positive comments.

Thank you for your consideration.

Quality of English Language: Extensive editing of English language required.

Thanks for your positive comments. As suggested, we have re-edited the whole article in English. We then have re-submitted the manuscript.

Thank you for your consideration.

Does the introduction provide sufficient background and include all relevant references? Can be improved.

Thanks for your positive comments. As suggested, we have revised and improved the introduction section. We then have re-submitted the manuscript.

Thank you for your consideration.

Is the research design appropriate? Can be improved.

Thanks for your positive comments. As suggested, we have improved the experiment design in the manuscript. We then have re-submitted the manuscript.

Thank you for your consideration.

Are the methods adequately described? Can be improved.

Thanks for your positive comments. As suggested, we have improved the descriptions of method in the manuscript in detail. We then have re-submitted the manuscript.

Thank you for your consideration.

Are the results clearly presented? Must be improved.

Thanks for your positive comments. As suggested, we have carefully revised and improved our results section in the manuscript. We then have re-submitted the manuscript.

Thank you for your consideration.

Are the conclusions supported by the results? Must be improved.

Thanks for your positive comments. As suggested, we have improved the conclusion section, namely: “In conclusion, under 100 mM Na2CO3 stress, 150 μM MT application elevated RWC, Pn, Gs, Tr, and Chl content, while reduced REC, Ci, and H2O2 content of leaves in maize seedlings. Meanwhile, the six candidate genes related to photosynthetic pigments and antioxidant homeostasis were up-regulated. They then formed complex interaction relationships to enhance salt tolerance in maize seedlings. Briefly, 150 μM MT application maintains the integrity of photosynthetic organs, scavenging excess ROS, and improving photosynthetic characteristics, subsequently alleviating damage and promoting the growth of maize seedlings under Na2CO3 environment (Figure 9).” In Lines 379-386 of the manuscript. We then have re-submitted the manuscript.

Thank you for your consideration.

Reviewer 2

Comments and Suggestions for Authors

The manuscript entitled, "Exogenous Melatonin Reinforces Photosynthesis, Antioxidant Defense, Expression of Stress-responsive Genes, and Ameliorates the Effects of Na2CO3 Stress in Maize" is a good study. The experimental design is not correct. There should be three factors, i.e., a) maize genotypes (P-138 and EMS-104) b) sowing conditions (non-saline and saline), and c) growth promoter application (without melatonin and without melatonin).

Thanks for your positive comments. As suggested, we Improve our experimental design were that “The seeds of P138 and EMS-104 were sterilized with 0.5% (v/v) sodium hypochlorite solution for 15 min, rinsed five times with ddH2O, and then soaked in ddH2O for 24 h at 22±0.5°C environment. Subsequently the ten soaked seeds were planted in sterilized vermiculite plastic boxes (13 cm diameter × 11 cm high), they then cultured in a greenhouse (25±0.5°C temperature; 300 μM · m-2 s-1 light intensity; 65% relative humidity; 12/12 h light/dark) for 15 d. The three-leaf seedlings were then treated four treatments for 7 d, i.e., 0 mM Na2CO3+0 μM MT (ddH2O, CK), 100 mM Na2CO3+0 μM MT(N), 0 mM Na2CO3+150 μM MT(M), and 100 mM Na2CO3+150 μM MT(MN). During the all treatments, the seedlings was washed 50 mL above corresponding mixed solution every two days. Each treatment was repeated three times.” In Lines 404-413 of the manuscript. We then have re-submitted the manuscript.

Thank you for your consideration.

  1. Add the p values in the figures as well. Revise the manuscript accordingly and submit it again.

Thank you for your suggestion. In our paper, we added the P-value in the note of the diagram, if we add it to the diagram again, we think it will be repetitive

Thank you for your consideration.

Abstract

It is well written. however, the novelty is not clearly explained.

Thanks for your positive comments. The innovation of our experiment is the following three points; First of all, the experimental materials used are P138 (P138 is a representative inbred line, derives from the P group, with weak salt tolerant) and EMS-104 (After treating P138 pollen with ethyl methane sulfonate solution (EMS), EMS-104 mutant with high salt tolerance was found from 1041 maize mutants), which provides valuable genetic resources for salt tolerance breeding in future. Secondly, P138 and EMS-104 were selected to explore the regulatory mechanism of MT on Na2CO3. Finally, we explored the regulatory mechanism of MT on Na2CO3 from the Growth Phenotypes (seedling weight, root weight), seedling length, and leaf area), chlorophyll content (chlorophyll a, chlorophyll b, carotenoids, and chlorophyll a/b), Antioxidant enzyme system (superoxide dismutase (SOD) activity, peroxidase (POD) activity, catalase (CAT) activity, ascorbate peroxidase (APX) activity), Stomata(paraxial surface stomatal length, dorsal surface stomatal length, paraxial surface stomatal width, dorsal surface stomatal width, paraxial surface stomatal area, and dorsal surface stomatal area), Genes Expression(superoxide dismutase (SOD; Zm00001d025106, Zm00001d031908), catalase 2 (CAT2; Zm00001d027511), peroxidase 72 (POD72; Zm00001d040364), chlorophyllide a oxygenase chloroplastic (Zm00001d011819), and nine-cis-epoxycarotenoid dioxygenase8 (Zm00001d017766)), and Photosynthetic Characteristics (net photosynthetic rate, stomatal conductance, intercellular CO2 concentration, and transpiration rate), respectively. According to the above three aspects, we improve and modify the abstract part were that “Salt stress can seriously affect the growth and development of maize (Zea mays L.), resulting in a great yield loss. Melatonin (MT), an indole hormone, as a potential enhancer of plant tolerance against salt stress. While the complex mechanisms of MT application enhancing maize salt tolerance still unclear. Herein, the three-leaf seedlings of salt-susceptible P138 and its salt-resistant ethyl methane sulfonate (EMS)-104 mutant, were cultured with or without 150 μM MT application under 0 and 100 mM Na2CO3 treatments for seven days, to systematically explore the response mechanisms of exogenous MT improving salt tolerance of maize. The results showed that salt stress triggered an escalation in reactive oxygen species production, enhanced multiple antioxidant enzymes activities, compromised cellular membrane permeability, inhibited photosynthetic pigments accumulation, and ultimately undermined the vigor and photosynthetic prowess of the seedlings. While the suitable MT application counteracted the detrimental impacts of Na2CO3 on seedlings growth and photosynthetic capacity, the seedling length and net photosynthetic rate of P138 and EMS-104 were increased by 5.5%and 18.7%, 12.7% and 54.5%, respectively. The quantitative real-time PCR (qRT-PCR) analysis further showed that MT application activated the expression levels of antioxidant enzymes-related genes (Zm00001d025106, Zm00001d031908, Zm00001d027511, and Zm00001d040364) and pigment biosynthetic genes (Zm00001d011819 and Zm00001d017766) in both maize seedlings under Na2CO3 stress, they then formed a complex interaction networks among genes expression, multiple physiological metabolism, and phenotype changes to influence salt tolerance of maize seedlings under MT or Na2CO3 stress. To sum up, these observations underscore that the 150 μM MT can alleviate salt injury of maize seedlings, which may provide new insights to further investigate MT regulation mechanism on enhancement maize seedlings salt resistance.” in Lines 10-30 of the manuscript. We then have re-submitted the manuscript.

Thank you for your consideration.

Introduction

Write the saline area in the World and in China

Why is it increasing every year and how much is it increasing?

Overall, in the world the agriculture loss due to salinity in dollars  

Write the maize production and area in the World and in China

Thanks for your positive comments. As suggested, we have revised the corresponding contents were that “Globally, saline-alkali soil comprises approximately 9.54 × 108 ha, with China alone possessing 9.91 × 107 ha [24]. By 2050, it is expected that about 50% of agricultural land worldwide will be affected by salinity [4].” in Lines 36-38 of the manuscript. We then have re-submitted the manuscript.

Thank you for your consideration.

Novelty is not clearly explained in the objectives.

Thanks for your positive comments. As suggested, we have revised the corresponding content were that “Considering the crucial importance of MT in boosting plant resistance. In this investigation, the salt-susceptible maize inbred line P138 and its salt-resistant EMS-104 (from a library of ethyl methane sulfonate (EMS)-induced mutants) mutant were used as materials, their seedlings at three-leaf stage were treated with or without 150 μM MT application under 0 and 100 mM Na2CO3 treatments for seven days to analyze seedlings growth status, photosynthetic pigments accumulation, photosynthetic ability, ROS levels, and antioxidant enzymes activities; meanwhile, the expression levels of six candidate genes responsible for antioxidant enzymes and photosynthesis in both genotypes seedlings under all treatments were also quantitatively analyzed by quantitative real-time PCR (qRT-PCR). Thereby, this study will help us determine the mechanism by which MT alleviates the impact of salt stress on maize seedlings, and gather insights into the mechanisms of salt tolerance in maize.” in Lines 71-82 of the manuscript. We then have re-submitted the manuscript.

Thank you for your consideration.

Overall, try to cite new studies and remove old.

Thanks for your positive comments. As suggested, we replace old references with more recent references. We then have re-submitted the manuscript.

Thank you for your consideration.

Results

In figure 1: the spelling of seedling length is not correct, correct it.

Thanks for your positive comments. As suggested, we changed the name of the seedling legend in Figure 1. we have revised and improved the manuscript. We then have re-submitted the manuscript.

Thank you for your consideration.

Figure 1. Changes of seedling weight (A), root weight (B), seedling length (C), and leaf area (D) of P138 and EMS-104 seedlings under different treatments, including 0 mM Na2CO3+0 μM melatonin (MT) treatment (CK); 100 mM Na2CO3+0 μM MT treatment (N); 0 mM Na2CO3+150 μM MT treatment (M); 100 mM Na2CO3+150 μM MT treatment (MN). Different lowercase letters indicated significant differences in p < 0.05 level.

Discussion

Cite the figure numbers in this section

Thanks for your positive comments. As suggested, we refer to the figures in each subsection in the discussion. We then have re-submitted the manuscript.

Thank you for your consideration.

More comparative and recent works can be cited. Remove old references.

Thanks for your comments, we have replaced the old reference with the new one. We then have re-submitted the manuscript.

Thank you for your consideration.

Materials and Methods

What is the dimension of boxes “plastic boxes”? What kind of material have you added in these boxes? Sand, soil etc?

Thank you for your questions, the size of the "plastic box" is (13 cm diameter x 11 cm high), and in the plastic box, we added vermiculite that has been sterilized. We then added content and resubmitted the manuscript.

Thank you for your consideration.

Comments on the Quality of English Language

Moderate editing of English language required.

Thanks for your positive comments. As suggested, we have re-edited the whole article in English. We then have re-submitted the manuscript.

Thank you for your consideration.

Open Review: I would not like to sign my review report.

Thanks for your positive comments.

Thank you for your consideration.

Quality of English Language: Moderate editing of English language required.

Thanks for your positive comments. As suggested, we have re-edited the whole article in English. We then have re-submitted the manuscript.

Thank you for your consideration.

Does the introduction provide sufficient background and include all relevant references? Must be improved.

Thanks for your positive comments. As suggested, we have revised and improved the introduction section. We then have re-submitted the manuscript.

Thank you for your consideration.

Is the research design appropriate? Must be improved.

Thanks for your positive comments. As suggested, we have improved the experiment design in the manuscript. We then have re-submitted the manuscript.

Thank you for your consideration.

Are the methods adequately described? Must be improved.

Thanks for your positive comments. As suggested, we have improved the descriptions of method in the manuscript in detail. We then have re-submitted the manuscript.

Thank you for your consideration.

Are the results clearly presented? Must be improved.

Thanks for your positive comments. As suggested, we have carefully revised and improved our results section in the manuscript. We then have re-submitted the manuscript.

Thank you for your consideration.

Are the conclusions supported by the results? Must be improved.

Thanks for your positive comments. As suggested, we have improved the conclusion section, namely: “In conclusion, under 100 mM Na2CO3 stress, 150 μM MT application elevated RWC, Pn, Gs, Tr, and Chl content, while reduced REC, Ci, and H2O2 content of leaves in maize seedlings. Meanwhile, the six candidate genes related to photosynthetic pigments and antioxidant homeostasis were up-regulated. They then formed complex interaction relationships to enhance salt tolerance in maize seedlings. Briefly, 150 μM MT application maintains the integrity of photosynthetic organs, scavenging excess ROS, and improving photosynthetic characteristics, subsequently alleviating damage and promoting the growth of maize seedlings under Na2CO3 environment (Figure 9).” In Lines 379-386 of the manuscript. We then have re-submitted the manuscript.

Thank you for your consideration.

Reviewer 3

Comments and Suggestions for Authors

The manuscript entitled "Exogenous Melatonin Reinforces Photosynthesis, Antioxidant Defense, Expression of Stress-responsive Genes, and Amelio4 rates the Effects of Na2CO3 Stress in Maize" aims to detect the effects of "MT in boosting plant resilience, in this investigation, we employed the salt-susceptible maize inbred line P138 and its salt-resistant derivative EMS-104 (from a library of ethyl methane sulfonate (EMS)-induced mutants) as experimental subjects." Although the manuscript has potential, the current version lacks specific methodological details, which prevents the reader to understand the design and what was done, as well as any possible replication of this study. For instance:

Thanks for your positive comments. As suggested, we add the corresponding details. We then re-submitted the manuscript.

Thank you for your consideration.

  1. how many biological replicates were used?

Thank you for your questions, our experiment set up three biological replicates. We then added content and resubmitted the manuscript.

Thank you for your consideration.

  1. What are the characteristics of the lines/mutants used?

Thank you for your questions, as suggested, we have revised the corresponding contents were that “P138 is a representative inbred line, derives from the P group, with weak salt tolerant [68].” in Lines 417-418 of the manuscript. We then added content and resubmitted the manuscript.

Thank you for your consideration.

  1. Plants were said to growth under greenhouse conditions? - thus, how were other factors controlled in order to not have any effect on results?

Thank you for your questions, All the materials in our experiment were planted in the same batch, except for the different mixed solution added during processing, and other conditions were consistent. We redescribed the specific content were that “The seeds of P138 and EMS-104 were sterilized with 0.5% (v/v) sodium hypochlorite solution for 15 min, rinsed five times with ddH2O, and then soaked in ddH2O for 24 h at 22±0.5°C environment. Subsequently the ten soaked seeds were planted in sterilized vermiculite plastic boxes (13 cm diameter × 11 cm high), they then cultured in a greenhouse (25±0.5°C temperature; 300 μM · m-2 s-1 light intensity; 65% relative humidity; 12/12 h light/dark) for 15 d, During this period, the seedlings was washed 50 mL ddH2O every two days. The three-leaf seedlings were then treated four treatments for 7 d, i.e., 0 mM Na2CO3+0 μM MT (ddH2O, CK), 100 mM Na2CO3+0 μM MT(N), 0 mM Na2CO3+150 μM MT(M), and 100 mM Na2CO3+150 μM MT(MN). During the all treatments, the seedlings was washed 50 mL above corresponding mixed solution every two days. Each treatment was repeated three times.” In Lines 404-414 of the manuscript. we then added content and resubmitted the manuscript.

Thank you for your consideration.

  1. Three seedlings were used? What is meant by seedlings here? In addition, why only 3? - this is a very low number.

Thank you for your questions, the three seedlings here refer to three replicates randomly selected under each treatment, and we actually planted 10 seeds per treatment per pot. In addition, we have refined our experimental design are that “The seeds of P138 and EMS-104 were sterilized with 0.5% (v/v) sodium hypochlorite solution for 15 min, rinsed five times with ddH2O, and then soaked in ddH2O for 24 h at 22±0.5°C environment. Subsequently the ten soaked seeds were planted in sterilized vermiculite plastic boxes (13 cm diameter × 11 cm high), they then cultured in a greenhouse (25±0.5°C temperature; 300 μM · m-2 s-1 light intensity; 65% relative humidity; 12/12 h light/dark) for 15 d, During this period, the seedlings was washed 50 mL ddH2O every two days. The three-leaf seedlings were then treated four treatments for 7 d, i.e., 0 mM Na2CO3+0 μM MT (ddH2O, CK), 100 mM Na2CO3+0 μM MT(N), 0 mM Na2CO3+150 μM MT(M), and 100 mM Na2CO3+150 μM MT(MN). During the all treatments, the seedlings was washed 50 mL above corresponding mixed solution every two days. Each treatment was repeated three times.” In Lines 404-414 of the manuscript. we then added content and resubmitted the manuscript.

Thank you for your consideration.

  1. "The four photosynthetic parameters, including Pn, Tr, Gs, Ci" - please describe the parameters and how they were measured.

Thank you for your questions, the definition of the four photosynthetic parameters and how they are measured.

1.Pn: Net Photosynthetic Rate refers to the amount of carbon dioxide fixed by photosynthesis of plant leaves in a unit time, usually expressed in μmol CO2 m^-2 s^-1.

2.Gs: Stomatal Conductance, indicating the openness of leaf stomata to gas exchange, usually expressed in mol m^-2 s^-1, which is closely related to the rate of photosynthesis and transpiration.

3.Ci: Intercellular CO2 Concentration refers to the concentration of carbon dioxide in the intercellular space of the leaves, usually expressed in μmol mol^-1 or ppm, reflecting the supply of CO2 inside the leaves.

4.Tr: Transpiration Rate refers to the amount of water vapor released by plant leaves through stomata per unit time, usually expressed as mmol H2O m^-2 s^-1, and is related to the water use efficiency of plants.

The four photosynthetic parameters, including net photosynthetic rate (Pn; μM CO2·m-2·s-1), intercellular CO2 (Ci; μM CO2·mol-1), stomatal conductance (Gs; M H2O·m-2·s-1), and transpiration rate (Tr; mM H2O·m-2·s-1) of the 3rd leaf in P138 and EMS-104 seedlings under all treatments was measured using a LI-6400XT (LI-CORIn, USA) portable photosynthetic measurement system. Set the illuminance inside the leaf chamber to 1000 μM/ (m2 · s), CO2 concentration of 400 μM · M-1, at the temperature of 25°C.

We filled in the details and resubmitted the manuscript

Thank you for your consideration.

  1. How many samples were used in rt-PCRs?

Thanks for your positive comments. As suggested, we have revised the corresponding contents were that “The total RNA was extracted from eight samples of the 3rd leaf of salt-sensitive P138 and salt-resistant EMS-104 seedlings (three biological replicates for each sample) under above four treatments with TRIZOL reagent (Invitrogen, USA), which was reverse-transcribed into cDNA using a SuperScript III First strand Kit (Invitrogen, Gibco). The qRT-PCR was conducted using TransStart Tip Green qPCR SuperMix (Tran, Beijing, China). Primers (Table 1) for six candidate genes involved in antioxidant enzymes activities and photosynthetic pigments biosynthesis were designed via Primer3web v.4.1.0 (https://primer3.ut.ee/; accessed on 13 May 2024). The positions of these genes were mapped in the Zea mays B73_V4 reference genome (https://www.maizegdb.org/; accessed on 13 May 2024), and their functional annotations were performed using the tool AgBase v2.00 (https://agbase.arizona.edu/; accessed on 13 May 2024). Relative gene expression level was calculated by the 2−∆∆Ct method, with Zm00001d010159 as an internal reference gene.” in Lines 468-479 of the manuscript. We then have re-submitted the manuscript.

Thank you for your consideration.

  1. An ANOVA and Duncan test were used. Which were the factors included, the dependent and independent variables?

Thanks for your positive comments. As suggested, we have revised the corresponding contents were that “For all tested traits of the two maize genotypes under all treatments, their one-way ANOVA was performed using the BM-SPSS Statistics v.20.0 software (SPSS, Chicago, IL., USA; https://www.Ibm.com/products/spss-statistics; accessed on June 16 2024). Pearson correlation among all traits were analyzed the Origin 2021(v. 21.0, OriginPro, USA; https://www.genescloud.cn; accessed on June 16 2024).” in Lines 483-487 of the manuscript. We then have re-submitted the manuscript.

Thank you for your consideration.

  1. These are only an example of the details missing in this document. In addition, the discussion is very poor.

Thanks for your positive comments. As suggested, we added the missing details in the article, and rewrote and improved the discussion section. We then have re-submitted the manuscript.

Thank you for your consideration.

  1. Proofreading comments: many sentences are incomplete, have typos and grammatical inconsistencies.

Thanks for your comments. As suggested, we have revised and supplemented what we found about incomplete sentences, typos and grammatical problems. We then have re-submitted the manuscript.

Thank you for your consideration.

Open Review: I would not like to sign my review report.

Thanks for your positive comments.

Thank you for your consideration.

Quality of English Language: Moderate editing of English language required.

Thanks for your positive comments. As suggested, we have re-edited the whole article in English. We then have re-submitted the manuscript.

Thank you for your consideration.

Does the introduction provide sufficient background and include all relevant references? Yes.

Thanks for your positive comments.

Thank you for your consideration.

Is the research design appropriate? Must be improved.

Thanks for your positive comments. As suggested, we have improved the experiment design in the manuscript. We then have re-submitted the manuscript.

Thank you for your consideration.

Are the methods adequately described? Must be improved.

Thanks for your positive comments. As suggested, we have improved the descriptions of method in the manuscript in detail. We then have re-submitted the manuscript.

Thank you for your consideration.

Are the results clearly presented? Can be improved.

Thanks for your positive comments. As suggested, we have carefully revised and improved our results section in the manuscript. We then have re-submitted the manuscript.

Thank you for your consideration.

Are the conclusions supported by the results? Can be improved.

Thanks for your positive comments. As suggested, we have improved the conclusion section, namely: “In conclusion, under 100 mM Na2CO3 stress, 150 μM MT application elevated RWC, Pn, Gs, Tr, and Chl content, while reduced REC, Ci, and H2O2 content of leaves in maize seedlings. Meanwhile, the six candidate genes related to photosynthetic pigments and antioxidant homeostasis were up-regulated. They then formed complex interaction relationships to enhance salt tolerance in maize seedlings. Briefly, 150 μM MT application maintains the integrity of photosynthetic organs, scavenging excess ROS, and improving photosynthetic characteristics, subsequently alleviating damage and promoting the growth of maize seedlings under Na2CO3 environment (Figure 9).” In Lines 379-386 of the manuscript. We then have re-submitted the manuscript.

Thank you for your consideration.

Sincerely,

Xiaoqiang Zhao professor

State Key Laboratory of Aridland Crop Science, Gansu Agricultural University

E-mail: zhaoxq3324@163.com

Reviewer 2 Report

Comments and Suggestions for Authors

The manuscript entitled, "Exogenous Melatonin Reinforces Photosynthesis, Antioxidant Defense, Expression of Stress-responsive Genes, and Ameliorates the Effects of Na2CO3 Stress in Maize" is a good study. The experimental design is not correct. There should be three factors, i.e., a) maize genotypes (P-138 and EMS-104) b) sowing conditions (non-saline and saline), and c) growth promoter application (without melatonin and without melatonin). Add the p values in the figures as well. Revise the manuscript accordingly and submit it again.

Abstract

·         It is well written, however, the novelty is not clearly explained.

Introduction

·         Write the saline area in the World and in China

·         Why is it increasing every year and how much is it increasing?

·         Overall, in the world the agriculture loss due to salinity in dollars

·         Write the maize production and area in the World and in China

·         Novelty is not clearly explained in the objectives.

·         Overall, try to cite new studies and remove old.

Results

·         In figure 1: the spelling of seedling length is not correct, correct it.

Discussion

·         Cite the figure numbers in this section

·         More comparative and recent works can be cited. Remove old references.

Materials and Methods

·         What is the dimension of boxes “plastic boxes”? What kind of material have you added in these boxes? Sand, soil etc?

Comments on the Quality of English Language

Moderate editing of English language required.

Author Response

(The authors gave the same response as above.)

Reviewer 3 Report

Comments and Suggestions for Authors

The manuscript entitled "Exogenous Melatonin Reinforces Photosynthesis, Antioxidant Defense, Expression of Stress-responsive Genes, and Amelio4 rates the Effects of Na2CO3 Stress in Maize" aims to detect the effects of "MT in boosting plant resilience, in this investigation, we employed the salt-susceptible maize inbred line P138 and its salt-resistant derivative EMS-104 (from a library of ethyl methane sulfonate (EMS)-induced mutants) as experimental subjects." Although the manuscript has potential, the current version lacks specific methodological details, which prevents the reader to understand the design and what was done, as well as any possible replication of this study. For instance:

- how many biological replicates were used?

- What are the characteristics of the lines/mutants used?

- Plants were said to growth under greenhouse conditions? - thus, how were other factors controlled in order to not have any effect on results?

- Three seedlings were used? What is meant by seedlings here? In addition, why only 3? - this is a very low number.

- "The four photosynthetic parameters, including Pn, Tr, Gs, Ci" - please describe the parameters and how they were measured.

- How many samples were used in rt-PCRs?

- An ANOVA and Duncan test were used. Which were the factors included, the dependent and independent variables?

These are only an example of the details missing in this document. In addition, the discussion is very poor. 

Proofreading comments: many sentences are incomplete, have typos and grammatical inconsistencies. 

Comments on the Quality of English Language

See above.

Author Response

(The authors gave the same response as above.)

Round 2

Reviewer 1 Report

Comments and Suggestions for Authors

The paper has been improved. However, some revisions needed to be done before publication.

 1. The lines were not numbered or were numbered incorrectly thus it is difficult to understand what the authors are referring to in the text in their responses.

2. P. 2.

“Considering the crucial importance of MT in boosting plant resistance.”

The sentence seems to be unfinished.

3. Although the language has been improved there are still some deficiencies.

For instance, P. 9:

“Phenotypes were directly observed that salt stress negatively affected the biomass and plant height of plants [32–34].”

This sentence was written incorrectly.

P. 10.

“For other exogenous hormones application, similar results were confirmed that exogenous salicylic acid (SA) increased the chlorophyll content of maize under salt stress [45].”

Passive voice should not be used here.

P. 10.

“The changes of stomatal morphology are positive physiological responses on different environmental conditions [47].”

This sentence is awkward.

I encourage the authors to inspect the text again to improve the language.

4. The comment 8 from 1st round of review was not actually addressed.

“In addition, some studies have shown that there is a positive feedback relationship between the content of chlorophyll and ROS metabolism, the increase of chlorophyll levels helped to scavenge ROS accumulation in plant [46].”

Again, this statement is questionable.

“…the increase of chlorophyll levels helped to scavenge ROS accumulation in plant”?

Is this possible? No mention of chlorophyll in the reference by Li et al. (46).

5. P. 10.

“The changes of stomatal morphology are positive physiological responses on different environmental conditions [47]. Some studies have shown that salt stress led to stomatal closure, and exogenous Si treatment improved the Gs of alfalfa (Medicago sativa L.) seedlings under salt stress [48]. Similarly, exogenous SA application improved the Gs of barley under salt stress [49].”

The references (47-49) in this paragraph are incorrect. What does Si mean?

6. P. 11. 4.1 Materials.

“Previous…” should be replaced by “Previously…”

“…with weak salt tolerant [62].” should be replaced by “…with weak salt tolerance [62].”

7. MM section.

The comment 16 from 1st round of review was not fully addressed.

There are places where the imperative mood has remained. For instance: Subsection 4.4.

8. MM section, subsection 4.8.

“The 1.0 g of fresh leaves of P138 and EMS-104 seedlings under all treatments, soaking them in 10 mL of 95% alcohol for 48 h, then the absorbance value of leach liquor was measured using a multifunctional enzyme marker model (SynergyHTX, USA) at 665 nm, 649 nm, and 470 nm.”

The sentence seems incorrect. Please rewrite.

Comments on the Quality of English Language

There are still some deficiencies. The languauge must be improved.

Author Response

Thank you for your letter of – and for the referee’s comments concerning our manuscript, “Exogenous Melatonin Reinforces Photosynthesis, Antioxidant Defense, and Their Genes Expression to Ameliorate Na2CO3 Stress in Maize (Manuscript ID: plants-3194664)”. We have carefully studied these comments and have made corresponding corrections to the manuscript, which we describe in detail below. We would like to re-submit the manuscript and that for possible publication on the Special Issue: “Effect of Biotic and Abiotic Factors on the Physiology of Horticultural Plants” of Plants. Thank you very much for your time and consideration.

Editor:

Your manuscript has now been reviewed by experts in the field and can be found with the review reports at: https://susy.mdpi.com/user/manuscripts/resubmit/fa18f33c95d35dc6f3ee60b75add3ffe Please revise the manuscript found at the above link according to the reviewers' comments and upload the revised file within 10 days.

Thanks for the positive comments of you and all reviewers for our manuscript. As suggested, we have carefully revised and improved our manuscript using the “Track Changes” function of the manuscript at the above link. We then have re-submitted the manuscript within the allotted time.

Thank you for your consideration.

(I) Ensure all references are relevant to the content of the manuscript.

Thanks for the positive comments. As suggested, we have carefully checked all references. We then have re-submitted the manuscript.

Thank you for your consideration.

(II) Highlight any revisions to the manuscript, so editors and reviewers can see any changes made.

Thanks for the positive comments. As suggested, we have carefully revised and improved our manuscript using the “Track Changes” function of the manuscript. We then have re-submitted the manuscript.

Thank you for your consideration.

(III) Provide a cover letter to respond to the reviewers’ comments and explain, point by point, the details of the manuscript revisions.

Thanks for your positive comments for our manuscript. As suggested, we have carefully revised and improved our manuscript. In addition, we have prepared a detailed response letter to all reviewers for each point, and then have re-submitted the manuscript.

Thank you for your consideration.

(IV) If the reviewer(s) recommended references, critically analyze them to ensure that their inclusion would enhance your manuscript. If you believe these references are unnecessary, you should not include them.

Thanks for your positive comments for our manuscript. As suggested, we have carefully checked and revised the References. At the same time, we also have re-added twenty-one new references to enhance the quality of our manuscript. We then have re-submitted the manuscript.

Thank you for your consideration.

 (V) If you found it impossible to address certain comments in the review reports, include an explanation in your appeal.

Thanks for your positive comments for our manuscript. As suggested, we have carefully revised and improved our manuscript. In addition, we have prepared a detailed response letter to all reviewers for each point, and then have re-submitted the manuscript.

Thank you for your consideration.

If your manuscript requires improvement to the language and/or figures, you may consider MDPI Author Services: https://www.mdpi.com/authors/english. Please note the status of this invitation “Publish Author Biography on the webpage of the paper” - https://susy.mdpi.com/user/manuscript/author_biography/b0aa56b7bef0a99dd4fe8c1bf66a47d6. If you wish to publish your biography, please complete it before your manuscript is accepted.

Thanks for the positive comments. As suggested, we have carefully checked and revised the English language of the manuscript. We then re-submitted the manuscript.

In addition, thanks for your invitation, we decided not to publish our biography.

Thank you for your consideration.

Please do not hesitate to contact us if you have any questions regarding the revision of your manuscript or if you need more time. We look forward to hearing from you soon.

Thanks for your positive comments for our manuscript. As suggested, we have carefully revised and improved the manuscript using the “Track Changes” function of our manuscript at the above link. We then have re-submitted the manuscript within the allotted time.

Thank you for your consideration.

Reviewer 1

Comments and Suggestions for Authors

The paper has been improved. However, some revisions needed to be done before publication.

Thanks for your positive comments. As suggested, we modified the corresponding content. We then re-submitted the manuscript.

Thank you for your consideration.

  1. The lines were not numbered or were numbered incorrectly thus it is difficult to understand what the authors are referring to in the text in their responses.

Thanks for your comments. We are very sorry for the inconvenience. As suggested, we have carefully corrected the page numbers and lines involved in this reply. Hope it will be convenient for you to comment.

Thank you for your consideration.

  1. P. 2. “Considering the crucial importance of MT in boosting plant resistance.” The sentence seems to be unfinished.

Thanks for your positive comments. As suggested, we added the corresponding content were that “Considering the crucial importance of MT in boosting plant resistance, it is of great significance to reveal the exogenous MT application for the mitigation mechanism of maize seedlings under salt stress.” in Lines 72-74 on page 2 of the manuscript. We then have re-submitted the manuscript.

Thank you for your consideration.

  1. Although the language has been improved there are still some deficiencies.

Thanks for your positive comments. As suggested, we improved on the deficiencies in the language. We then have re-submitted the manuscript.

Thank you for your consideration.

  1. For instance, P. 9: “Phenotypes were directly observed that salt stress negatively affected the biomass and plant height of plants [32–34].” This sentence was written incorrectly.

Thanks for your positive comments. As suggested, we modified the corresponding content were that “Many studies were clearly observed that the phenotypes of plants under salt stress exhibited unfavourable effects, especailly in biomass and plant height [32–34].” in Lines 327-328 on page 11 of the manuscript. We then have re-submitted the manuscript.

Thank you for your consideration.

  1. P. 10. “For other exogenous hormones application, similar results were confirmed that exogenous salicylic acid (SA) increased the chlorophyll content of maize under salt stress [45].” Passive voice should not be used here.

Thanks for your positive comments. As suggested, we modified the corresponding content were that “For other exogenous hormones application, similar results confirmed that exogenous salicylic acid (SA) increased the chlorophyll content of maize under salt stress [45].” in Lines 356-357 on page 12 of the manuscript. We then have re-submitted the manuscript.

Thank you for your consideration.

  1. P. 10. “The changes of stomatal morphology are positive physiological responses on different environmental conditions [47].” This sentence is awkward.

Thanks for your positive comments. As suggested, we modified the corresponding content were that “The changes of stomatal morphology can positive reflect the physiological changes of plants under different environmental conditions [47].” in Lines 373-374 on page 12 of the manuscript. We then have re-submitted the manuscript.

Thank you for your consideration.

  1. I encourage the authors to inspect the text again to improve the language.

Thanks for your positive comments. As suggested, we rechecked the text and improved the language. We then have re-submitted the manuscript.

Thank you for your consideration.

  1. The comment 8 from 1stround of review was not actually addressed. “In addition, some studies have shown that there is a positive feedback relationship between the content of chlorophyll and ROS metabolism, the increase of chlorophyll levels helped to scavenge ROS accumulation in plant [46].”

Again, this statement is questionable. “…the increase of chlorophyll levels helped to scavenge ROS accumulation in plant”? Is this possible? No mention of chlorophyll in the reference by Li et al. (46).

Thanks for your positive comments. As suggested, we have revised the corresponding contents were that “In addition, some studies have shown that appropriate ROS levels are beneficial for chloroplast development and function [46].” in Lines 366-367 on page 12 of the manuscript. We then have re-submitted the manuscript.

Thank you for your consideration.

  1. “The changes of stomatal morphology are positive physiological responses on different environmental conditions [47]. Some studies have shown that salt stress led to stomatal closure, and exogenous Si treatment improved the Gs of alfalfa (Medicago sativa L.) seedlings under salt stress [48]. Similarly, exogenous SA application improved the Gs of barley under salt stress [49].” The references (47-49) in this paragraph are incorrect. What does Si mean?

Thanks for your positive comments. As suggested, we re-proofread the references cited in the paper, and in the article added the silicon (Si) in Lines 375 on page 12 of the manuscript, so that readers can read easily. We then have re-submitted the manuscript.

Thank you for your consideration.

  1. “Previous…”should be replaced by “Previously…” “…with weak salt tolerant [62].” should be replaced by“…with weak salt tolerance [62].”

Thanks for your positive comments. As suggested, we replace “Previous…” With “Previously…” and “…with and weak salt tolerant [62].” With “…with weak salt tolerance [62].”. We then have re-submitted the manuscript.

Thank you for your consideration.

  1. MM section.

The comment 16 from 1st round of review was not fully addressed. There are places where the imperative mood has remained. For instance: Subsection 4.4.

Thanks for your positive comments. As suggested, we modified the Subsection 4.4 content was that “The 3rd fresh leaf of P138 and EMS-104 seedlings under all treatments was cut middle of leaf (1cm × 1 cm), and were stuck on a slide coated with glue for 20 minutes. The leaves sections were then delicately detached using tweezers, leaving the temporary stomatal slides for collection. Stomatal morphology was examined utilizing an integrated forward and inverted fluorescence microscope (Revolve RVL-100-G, ECHO USA). For each treatment, five random fields of view were selected from the paraxial/dorsal surface. Stomatal length, width, and area were measured and statistically recorded under a 4x objective (image size 3226 × 3024).” in Lines 445-452 on page 14 of the manuscript. We then have re-submitted the manuscript.

Thank you for your consideration.

  1. MM section, subsection 4.8. “The 1.0 g of fresh leaves of P138 and EMS-104 seedlings under all treatments, soaking them in 10 mL of 95% alcohol for 48 h, then the absorbance value of leach liquor was measured using a multifunctional enzyme marker model (SynergyHTX, USA) at 665 nm, 649 nm, and 470 nm.” The sentence seems incorrect. Please rewrite.

Thanks for your positive comments. As suggested, we have revised the corresponding contents were that “The 1.0 g fresh leaves of P138 and EMS-104 seedlings under all treatments were cut and soaked in 10 mL of 95% alcohol for 48 h, which were then centrifuged at 12,000 rpm (Centrifuge 5425/5425 R; Eppendorf, Germany) for 10 min. The absorbance of the supernatant was measured utilizing a multifunctional enzyme marker model (SynergyHTX, USA), with readings taken at wavelengths of 665 nm, 649 nm, and 470 nm.” in Lines 475-479 on page 15 of the manuscript. We then have re-submitted the manuscript.

Thank you for your consideration.

Comments on the Quality of English Language

There are still some deficiencies. The language must be improved.

Thanks for your positive comments. As suggested, we have re-edited the whole article in English. We then have re-submitted the manuscript.

Thank you for your consideration.

Open Review: I would not like to sign my review report.

Thanks for your positive comments.

Thank you for your consideration.

Quality of English Language: Moderate editing of English language required.

Thanks for your positive comments. As suggested, we have re-edited the whole article in English. We then have re-submitted the manuscript.

Thank you for your consideration.

Reviewer 2

Comments and Suggestions for Authors

Authors have tried to address some comments but again the statistical analysis is not corrected, your design should be CRD factorial (three way) as three factors, i.e., a) maize genotypes (P-138 and EMS-104) b), sowing conditions (non-saline and saline), and c) growth promoter application (without melatonin and without melatonin). Mention the p values in figures. You can take help from this article titled “Plant growth promoters boost the photosynthesis related mechanisms and secondary metabolism of late-sown wheat under contrasting saline regimes”.

Thanks for your positive comments. As suggested, we have obtained help from the article entitled “Plant growth promoters boost the photosynthesis related mechanisms and secondary metabolism of late-sown wheat under contrasting saline regimes” understood your suggestions, and made improvements after referring to this paper. In order not to affect the beauty of the figure, we reflected the P-value in the Results and Analysis 2.1 chapter. We then have re-submitted the manuscript.

Thank you for your consideration.

Results: The sole and interaction significance effect of maize genotypes, sowing conditions, and melatonin is not discussed.

Thanks for your positive comments. As suggested, we supplemented the results and analyses of the single and interactive significant effects of maize genotype, seeding conditions and melatonin were that “2.1. Combined Analysis of among All Traits

Multiple factorial analysis of variance (ANOVA) proved significant for the tested variables (including maize genotype (G), MT (M) application, and Na2CO3 (N) stress) and their physiological and molecular interactions on the 25 tested traits including four growth parameters, eight photosynthetic performances, one membrane characteristics, one ROS levels, four antioxidant enzymes activities, one Cell osmotic pressure, and six stomatal morphology in two maize genotypes seedlings under different treatments (Table 1). The findings demonstrated that the resistance to salt stress was controlled by the factor of maize’s own genetic constitution, exogenous MT application, Na2CO3 stress, and their interaction effects. Our results also showed that the genotypes, MT application, and Na2CO3 stress imposed higher influences on these traits than their interactions.

Table 1. Multiple factorial analysis of variance (ANOVA) of the three examined variables (maize genotypes, Melatonin (MT) application, and Na2CO3 stress), their interactions on 25 tested traits in 2 maize genotypes seedlings under different treatments.

Trait

Genotypes (G)

MT application (M)

Na2CO3 stress (N)

G × M

Interaction

G × N

Interaction

M × N

Interaction

G × M × N Interaction

Growth parameter

SW

F=293.846

(p < 0.01)

F=11.613

(p < 0.05)

F=99.567

(p < 0.01)

F=0.003

(p > 0.05)

F=2.867

(p > 0.05)

F=0.626

(p > 0.05)

F=0.498

(p > 0.05)

RW

F=102.119

(p < 0.01)

F=8.094

(p < 0.05)

F=81.9

(p < 0.01)

F=0.322

(p > 0.05)

F=3.575

(p > 0.05)

F=1.358

(p > 0.05)

F=0.219

(p > 0.05)

SL

F=117.896

(p < 0.01)

F=18.999

(p < 0.01)

F=84.356

(p < 0.01)

F=5.095

(p < 0.05)

F=1.551

(p > 0.05)

F=4.254

(p > 0.05)

F=0.341

(p > 0.05)

SA

F=86.728

(p < 0.01)

F=16.071

(p < 0.01)

F=59.154

(p < 0.01)

F=0.822

(p > 0.05)

F=0.310

(p > 0.05)

F=5.426

(p < 0.05)

F=0.381

(p > 0.05)

Photosynthetic performance

Chl a

F=1526.223

(p < 0.01)

F=234.472

(p < 0.01)

F=921.329

(p < 0.01)

F=11.266

(p < 0.05)

F=15.303

(p < 0.05)

F=4.887

(p < 0.05)

F=8.161

(p < 0.05)

Chl b

F=881.143

(p < 0.01)

F=115.734

(p < 0.01)

F=318.597

(p < 0.01)

F=9.448

(p < 0.01)

F=13.545

(p < 0.01)

F=10.185

(p < 0.01)

F=3.864

(p > 0.05)

Car

F=10617.084

(p < 0.01)

F=398.44

(p < 0.01)

F=7020.877

(p < 0.01)

F=11.972

(p < 0.01)

F=2286.435

(p < 0.01)

F=95.874

(p < 0.01)

F=25.402

(p < 0.01)

Chl a/b

F=2231.112

(p < 0.01)

F=329.807

(p < 0.01)

F=1197.072

(p < 0.01)

F=3.212

(p > 0.05)

F=25.15

(p < 0.01)

F=10.771

(p < 0.01)

F=11.366

(p < 0.01)

Pn

F=1769.026

(p < 0.01)

F=114.864

(p < 0.01)

F=3185.756

(p < 0.01)

F=6.570

(p < 0.05)

F=46.34

(p < 0.01)

F=140.736

(p < 0.01)

F=39.46

(p < 0.01)

Ci

F=607.036

(p < 0.01)

F=148.467

(p < 0.01)

F=3686.901

(p < 0.01)

F=31.929

(p > 0.05)

F=0.051

(p < 0.01)

F=242.723

(p < 0.01)

F=15.881

(p < 0.01)

Gs

F=2046.216

(p < 0.01)

F=656.639

(p < 0.01)

F=3607.745

(p < 0.01)

F=5.205

(p < 0.05)

F=126.93

(p < 0.01)

F=915.369

(p < 0.01)

F=156.986

(p < 0.01)

Tr

F=10224.121

(p < 0.01)

F=10506.233

(p < 0.01)

F=72188.349

(p < 0.01)

F=444.097

(p < 0.01)

F=20.843

(p < 0.01)

F=5111.278

(p < 0.01)

F=2.361

(p > 0.05)

Membrane characteristics

REC

F=111.259

(p < 0.01)

F=11.732

(p < 0.05)

F=150.98

(p < 0.01)

F=2.978

(p > 0.05)

F=14.421

(p < 0.05)

F=9.006

(p > 0.05)

F=2.38

(p > 0.05)

ROS level

H2O2

F=0.973

(p > 0.05)

F=687.669

(p < 0.01)

F=1734.042

(p < 0.01)

F=1.590

(p > 0.05)

F=3.304

(p > 0.05)

F=254.762

(p < 0.01)

F=0.015

(p > 0.05)

Antioxidant enzymes activity

SOD

F=83.512

(p < 0.01)

F=117.514

(p < 0.01)

F=1093.202

(p < 0.01)

F=2.202

(p > 0.05)

F=1.121

(p > 0.05)

F=83.542

(p < 0.01)

F=42.627

(p < 0.01)

POD

F=40.6

(p < 0.01)

F=5.576

(p < 0.05)

F=96.68

(p < 0.01)

F=0.002

(p > 0.05)

F=2.089

(p > 0.05)

F=2.924

(p > 0.05)

F=0.424

(p > 0.05)

CAT

F=73.32

(p < 0.01)

F=66.286

(p < 0.01)

F=1342.286

(p < 0.01)

F=0.493

(p > 0.05)

F=66.286

(p < 0.01)

F=0.966

(p > 0.05)

F=22.778

(p < 0.01)

APX

F=89.308

(p < 0.01)

F=89.308

(p < 0.01)

F=1231.482

(p < 0.01)

F=5.511

(p < 0.05)

F=64.091

(p < 0.01)

F=39.946

(p < 0.01)

F=0.438

(p > 0.05)

Cell osmotic pressure

RWC

F=71.781

(p < 0.01)

F=13.459

(p < 0.05)

F=120.623

(p < 0.01)

F=5.105

(p < 0.05)

F=11.782

(p < 0.05)

F=10.607

(p < 0.05)

F=4.819

(p < 0.05)

stomatal morphology

PSL

F=2284.202

(p < 0.01)

F=92.877

(p < 0.01)

F=478.062

(p < 0.01)

F=5.864

(p < 0.05)

F=1.873

(p > 0.05)

F=2.111

(p > 0.05)

F=40.174

(p < 0.01)

DSL

F=409.549

(p < 0.01)

F=365.623

(p < 0.01)

F=269.314

(p < 0.01)

F=63.733

(p < 0.01)

F=94.369

(p < 0.01)

F=6.45

(p < 0.05)

F=2.092

(p > 0.05)

PSW

F=7.348

(p < 0.05)

F=32.125

(p < 0.01)

F=71.911

(p < 0.01)

F=0.131

(p > 0.05)

F=0.093

(p > 0.05)

F=4.791

(p < 0.05)

F=0.131

(p > 0.05)

DSW

F=2079.043

(p < 0.01)

F=278.294

(p < 0.01)

F=1382.348

(p < 0.01)

F=6.591

(p < 0.05)

F=2.25

(p > 0.05)

F=83.75

(p < 0.01)

F=46.181

(p < 0.01)

PSA

F=1665.708

(p < 0.01)

F=65.615

(p < 0.01)

F=431.921

(p < 0.01)

F=0.455

(p > 0.05)

F=0.178

(p > 0.05)

F=0.011

(p > 0.05)

F=4.818

(p < 0.05)

DSA

F=1264.801

(p < 0.01)

F=135.369

(p < 0.01)

F=290.6

(p < 0.01)

F=2.98

(p < 0.05)

F=5.316

(p < 0.05)

F=14.257

(p < 0.01)

F=13.409

(p < 0.01)

SW, seedling weight; RW, root weight; SL, seedling length; LA, leaf area; Chl a, chlorophyll a; Chl b, chlorophyll b; Car, carotenoids; Chl a/b, chlorophyll a/b; Pn, net photosynthetic rate; Ci, intercellular CO2 concentration; Gs, stomatal conductance; Tr, transpiration rate; REC, relative electrical conductivity; ROS, reactive oxygen species; H2O2, H2O2 content; SOD, superoxide dismutase activity; POD, peroxidase activity; CAT, catalase activity; APX, ascorbate peroxidase activity; RWC, relative water content; PSL, paraxial surface stomatal length; DSL, dorsal surface stomatal length; PSW, paraxial surface stomatal width; DSW, dorsal surface stomatal width; PSA, paraxial surface stomatal area; DSA, dorsal surface stomatal area;” In Lines 86-107 on page 2-4 the of manuscript. We then have re-submitted the manuscript.

Thank you for your consideration.

Comments on the Quality of English Language

Minor editing of English language required

Thanks for your positive comments. As suggested, we have re-edited the whole article in English. We then have re-submitted the manuscript.

Thank you for your consideration.

Open Review: I would not like to sign my review report.

Thanks for your positive comments.

Thank you for your consideration.

Quality of English Language: Minor editing of English language required.

Thanks for your positive comments. As suggested, we have re-edited the whole article in English. We then have re-submitted the manuscript.

Thank you for your consideration.

Does the introduction provide sufficient background and include all relevant references? Can be improved.

Thanks for your positive comments. As suggested, we have revised and improved the introduction section. We then have re-submitted the manuscript.

Thank you for your consideration.

Is the research design appropriate? Must be improved.

Thanks for your positive comments. As suggested, we have improved the experiment design in the manuscript. We then have re-submitted the manuscript.

Thank you for your consideration.

Are the methods adequately described? Can be improved.

Thanks for your positive comments. As suggested, we have improved the descriptions of method in the manuscript in detail. We then have re-submitted the manuscript.

Thank you for your consideration.

Are the results clearly presented? Must be improved.

Thanks for your positive comments. As suggested, we have carefully revised and improved our results section in the manuscript. We then have re-submitted the manuscript.

Thank you for your consideration.

Are the conclusions supported by the results? Can be improved.

Thanks for your positive comments. As suggested, we have improved the conclusion section, namely: “In conclusion, under 100 mM Na2CO3 stress, 150 μM MT application elevated RWC, Pn, Gs, Tr, and Chl content, while reduced REC, Ci, and H2O2 content of leaves in maize seedlings. Meanwhile, the six candidate genes related to photosynthetic pigments and antioxidant homeostasis were up-regulated. They then formed complex interaction relationships to enhance salt tolerance in maize seedlings. Briefly, 150 μM MT application maintains the integrity of photosynthetic organs, scavenging excess ROS, and improving photosynthetic characteristics, subsequently alleviating damage and promoting the growth of maize seedlings under Na2CO3 environment (Figure 9).” In Lines 403-410 on page 13 of the manuscript. We then have re-submitted the manuscript.

Thank you for your consideration.

Reviewer 3

Comments and Suggestions for Authors

The authors have included all suggestions and have erased previous doubts 

Thanks for your positive comments.

Thank you for your consideration.

Comments on the Quality of English Language

See above.

Thanks for your positive comments. As suggested, we have re-edited the whole article in English. We then have re-submitted the manuscript.

Thank you for your consideration.

Open Review: I would not like to sign my review report.

Thanks for your positive comments.

Thank you for your consideration.

Quality of English Language: Minor editing of English language required.

Thanks for your positive comments. As suggested, we have re-edited the whole article in English. We then have re-submitted the manuscript.

Thank you for your consideration.

Does the introduction provide sufficient background and include all relevant references? Yes.

Thanks for your positive comments.

Thank you for your consideration.

Is the research design appropriate? Yes.

Thanks for your positive comments.

Thank you for your consideration.

Are the methods adequately described? Yes.

Thanks for your positive comments.

Thank you for your consideration.

Are the results clearly presented? Yes.

Thanks for your positive comments.

Thank you for your consideration.

Are the conclusions supported by the results? Yes.

Thanks for your positive comments.

Thank you for your consideration.

Sincerely,

Xiaoqiang Zhao professor

State Key Laboratory of Aridland Crop Science, Gansu Agricultural University

E-mail: zhaoxq3324@163.com

Reviewer 2 Report

Comments and Suggestions for Authors

Authors have tried to address some comments but again the statistical analysis is not corrected, your design should be CRD factorial (three way) as three factors, i.e., a) maize genotypes (P-138 and EMS-104) b), sowing conditions (non-saline and saline), and c) growth promoter application (without melatonin and without melatonin). Mention the p values in figures. You can take help from this article titled “Plant growth promoters boost the photosynthesis related mechanisms and secondary metabolism of late-sown wheat under contrasting saline regimes”.

Results:

The sole and interaction significance effect of maize genotypes, sowing conditions, and melatonin is not discussed.

Comments on the Quality of English Language

Minor editing of English language required

Author Response

(The authors gave the same response as above.)

Reviewer 3 Report

Comments and Suggestions for Authors

The authors have included all suggestions and have erased previous doubts 

Comments on the Quality of English Language

See above.

Author Response

(The authors gave the same response as above.)

Round 3

Reviewer 2 Report

Comments and Suggestions for Authors

Again Authors have tried to address many comments but again the statistical analysis is not corrected, your design should be CRD factorial (three way) as three factors, i.e., a) maize genotypes (P-138 and EMS-104) b), sowing conditions (non-saline and saline), and c) growth promoter application (without melatonin and without melatonin). But in all figures lettering are not correct.

Author Response

Thank you for your letter of – and for the referee’s comments concerning our manuscript, “Exogenous Melatonin Reinforces Photosynthesis, Antioxidant Defense, and Their Genes Expression to Ameliorate Na2CO3 Stress in Maize (Manuscript ID: plants-3194664)”. We have carefully studied these comments and have made corresponding corrections to the manuscript, which we describe in detail below. We would like to re-submit the manuscript and that for possible publication on the Special Issue: “Effect of Biotic and Abiotic Factors on the Physiology of Horticultural Plants” of Plants. Thank you very much for your time and consideration.

Editor:

Your manuscript has now been reviewed by experts in the field and can be found with the review reports at: https://susy.mdpi.com/user/manuscripts/resubmit/fa18f33c95d35dc6f3ee60b75add3ffe Please revise the manuscript found at the above link according to the reviewers' comments and upload the revised file within 10 days.

Thanks for the positive comments of you and all reviewers for our manuscript. As suggested, we have carefully revised and improved our manuscript using the “Track Changes” function of the manuscript at the above link. We then have re-submitted the manuscript within the allotted time.

Thank you for your consideration.

(I) Ensure all references are relevant to the content of the manuscript.

Thanks for the positive comments. As suggested, we have carefully checked all references. We then have re-submitted the manuscript.

Thank you for your consideration.

(II) Highlight any revisions to the manuscript, so editors and reviewers can see any changes made.

Thanks for the positive comments. As suggested, we have carefully revised and improved our manuscript using the “Track Changes” function of the manuscript. We then have re-submitted the manuscript.

Thank you for your consideration.

(III) Provide a cover letter to respond to the reviewers’ comments and explain, point by point, the details of the manuscript revisions.

Thanks for your positive comments for our manuscript. As suggested, we have carefully revised and improved our manuscript. In addition, we have prepared a detailed response letter to all reviewers for each point, and then have re-submitted the manuscript.

Thank you for your consideration.

(IV) If the reviewer(s) recommended references, critically analyze them to ensure that their inclusion would enhance your manuscript. If you believe these references are unnecessary, you should not include them.

Thanks for your positive comments for our manuscript. As suggested, we have carefully checked and revised the References. At the same time, we also have re-added twenty-one new references to enhance the quality of our manuscript. We then have re-submitted the manuscript.

Thank you for your consideration.

 (V) If you found it impossible to address certain comments in the review reports, include an explanation in your appeal.

Thanks for your positive comments for our manuscript. As suggested, we have carefully revised and improved our manuscript. In addition, we have prepared a detailed response letter to all reviewers for each point, and then have re-submitted the manuscript.

Thank you for your consideration.

If your manuscript requires improvement to the language and/or figures, you may consider MDPI Author Services: https://www.mdpi.com/authors/english. Please note the status of this invitation “Publish Author Biography on the webpage of the paper” - https://susy.mdpi.com/user/manuscript/author_biography/b0aa56b7bef0a99dd4fe8c1bf66a47d6. If you wish to publish your biography, please complete it before your manuscript is accepted.

Thanks for the positive comments. As suggested, we have carefully checked and revised the English language of the manuscript. We then re-submitted the manuscript.

In addition, thanks for your invitation, we decided not to publish our biography.

Thank you for your consideration.

Please do not hesitate to contact us if you have any questions regarding the revision of your manuscript or if you need more time. We look forward to hearing from you soon.

Thanks for your positive comments for our manuscript. As suggested, we have carefully revised and improved the manuscript using the “Track Changes” function of our manuscript at the above link. We then have re-submitted the manuscript within the allotted time.

Thank you for your consideration.

Reviewer 2

Comments and Suggestions for Authors

Again Authors have tried to address many comments but again the statistical analysis is not corrected, your design should be CRD factorial (three way) as three factors, i.e., a) maize genotypes (P-138 and EMS-104) b), sowing conditions (non-saline and saline), and c) growth promoter application (without melatonin and without melatonin). But in all figures lettering are not correct.

Thanks for your positive comments. Yes, we agree with your views. In this study, there were three factors, i.e., Maize genotypes (P138 and EMS-104), Salt treatments (0 and 100 mM Na2CO3), and Growth promoter application (0 and 150 μM MT). Therefore, we should consider the one-way variance and their interaction variance among all traits when the data is analyzed.

As suggested, we have carefully read the reference: “Hafeez, M.B.; Ghaffar, A.; Zahra, N.; Ahmad, N.; Hussain, S.; Li, J. Plant growth promoters boost the photosynthesis related mechanisms and secondary metabolism of late-sown wheat under contrasting saline regimes. Plant Stress 2024, 12, 100480”. On the basis of results of the multiple factorial analysis of variance in the reference, we thus have re-made our Figures as follows:

Figure 1. Changes of seedling weight (A), root weight (B), seedling length (C), and leaf area (D) of P138 and EMS-104 seedlings under different treatments, including 0 mM Na2CO3+0 μM melatonin (MT) treatment (CK); 100 mM Na2CO3+0 μM MT treatment (N); 0 mM Na2CO3+150 μM MT treatment (M); 100 mM Na2CO3+150 μM MT treatment (MN). Different lowercase letters indicated significant differences in p < 0.05 level. ns; nonsignificant (p > 0.05) *; Significant (p ≤ 0.05) and **; Significant (p ≤ 0.01).

Figure 2. Changes of superoxide dismutase (SOD) activity (A), peroxidase (POD) activity (B), catalase (CAT) activity (C), ascorbate peroxidase (APX) activity (D), and H2O2 content (E) of P138 and EMS-104 seedlings under different treatments, including 0 mM Na2CO3+0 μM melatonin (MT) treatment (CK); 100 mM Na2CO3+0 μM MT treatment (N); 0 mM Na2CO3+150 μM MT treatment (M); 100 mM Na2CO3+150 μM MT treatment (MN). Different letters indicate significant differences when p < 0.05. ns; nonsignificant (p > 0.05) *; Significant (p ≤ 0.05) and **; Significant (p ≤ 0.01).

Figure 3. Changes of relative electrical conductivity (REC; A), and relative water content (RWC; B) of P138 and EMS-104 seedlings under different treatments, including 0 mM Na2CO3+0 μM melatonin (MT) treatment (CK); 100 mM Na2CO3+0 μM MT treatment (N); 0 mM Na2CO3+150 μM MT treatment (M); 100 mM Na2CO3+150 μM MT treatment (MN). Different lowercase letters indicated significant differences in p < 0.05 level. ns; nonsignificant (p > 0.05) *; Significant (p ≤ 0.05) and **; Significant (p ≤ 0.01).

Figure 4. Changes of net photosynthetic rate (A), stomatal conductance (B), intercellular CO2 concentration (C), and transpiration rate (D) of P138 and EMS-104 seedlings under different treatments, including 0 mM Na2CO3+0 μM melatonin (MT) treatment (CK); 100 mM Na2CO3+0 μM MT treatment (N); 0 mM Na2CO3+150 μM MT treatment (M); 100 mM Na2CO3+150 μM MT treatment (MN). Different lowercase letters indicated significant differences in p < 0.05 level. ns; nonsignificant (p > 0.05) *; Significant (p ≤ 0.05) and **; Significant (p ≤ 0.01).

Figure 5. Changes of chlorophyll a (Chl a; A), chlorophyll b (Chl b; B), carotenoids (Car; C), and chlorophyll a/b (Chl a/b; D) of P138 and EMS-104 seedlings under different treatments, including 0 mM Na2CO3+0 μM melatonin (MT) treatment (CK); 100 mM Na2CO3+0 μM MT treatment (N); 0 mM Na2CO3+150 μM MT treatment (M); 100 mM Na2CO3+150 μM MT treatment (MN). Different lowercase letters indicated significant differences in p < 0.05 level. ns; nonsignificant (p > 0.05) *; Significant (p ≤ 0.05) and **; Significant (p ≤ 0.01).

Figure 6. Changes of paraxial surface stomatal length (A), dorsal surface stomatal length (B), paraxial surface stomatal width (C), dorsal surface stomatal width (D), paraxial surface stomatal area (E), and dorsal surface stomatal area (F) o of P138 and EMS-104 seedlings under different treatments, including 0 mM Na2CO3+0 μM melatonin (MT) treatment (CK); 100 mM Na2CO3+0 μM MT treatment (N); 0 mM Na2CO3+150 μM MT treatment (M); 100 mM Na2CO3+150 μM MT treatment (MN); Different lowercase letters indicated significant differences in p < 0.05 level. ns; nonsignificant (p > 0.05) *; Significant (p ≤ 0.05) and **; Significant (p ≤ 0.01).

In addition, we have also referred the reference: “Zhao, X.Q.; Zhao, C.; Niu, Y.N.; Chao, W.; He, W.; Wang, Y.F.; Mao, T.T.; Bai, X.D. Understanding and comprehensive evaluation of cold resistance in the seedlings of multiple maize genotypes. Plants 2022, 11, 1881.” And combining our results of multiple factorial analysis of variance among all tested traits. We have provided the detailed data about the results in Supplementary Table S 1, namely:

Table S1. Multiple factorial analysis of variance (ANOVA) of the three examined variables (maize genotypes, Melatonin (MT) application, and Na2CO3 stress), their interactions on 25 tested traits in 2 maize genotypes seedlings under different treatments.

Trait

Genotypes (G)

MT application (M)

Na2CO3 stress (N)

G × M

Interaction

G × N

Interaction

M × N

Interaction

G × M × N Interaction

Growth parameter

SW

F=293.846

(p < 0.01)

F=11.613

(p < 0.05)

F=99.567

(p < 0.01)

F=0.003

(p > 0.05)

F=2.867

(p > 0.05)

F=0.626

(p > 0.05)

F=0.498

(p > 0.05)

RW

F=102.119

(p < 0.01)

F=8.094

(p < 0.05)

F=81.9

(p < 0.01)

F=0.322

(p > 0.05)

F=3.575

(p > 0.05)

F=1.358

(p > 0.05)

F=0.219

(p > 0.05)

SL

F=117.896

(p < 0.01)

F=18.999

(p < 0.01)

F=84.356

(p < 0.01)

F=5.095

(p < 0.05)

F=1.551

(p > 0.05)

F=4.254

(p > 0.05)

F=0.341

(p > 0.05)

SA

F=86.728

(p < 0.01)

F=16.071

(p < 0.01)

F=59.154

(p < 0.01)

F=0.822

(p > 0.05)

F=0.310

(p > 0.05)

F=5.426

(p < 0.05)

F=0.381

(p > 0.05)

Photosynthetic performance

Chl a

F=1526.223

(p < 0.01)

F=234.472

(p < 0.01)

F=921.329

(p < 0.01)

F=11.266

(p < 0.05)

F=15.303

(p < 0.05)

F=4.887

(p < 0.05)

F=8.161

(p < 0.05)

Chl b

F=881.143

(p < 0.01)

F=115.734

(p < 0.01)

F=318.597

(p < 0.01)

F=9.448

(p < 0.01)

F=13.545

(p < 0.01)

F=10.185

(p < 0.01)

F=3.864

(p > 0.05)

Car

F=10617.084

(p < 0.01)

F=398.44

(p < 0.01)

F=7020.877

(p < 0.01)

F=11.972

(p < 0.01)

F=2286.435

(p < 0.01)

F=95.874

(p < 0.01)

F=25.402

(p < 0.01)

Chl a/b

F=2231.112

(p < 0.01)

F=329.807

(p < 0.01)

F=1197.072

(p < 0.01)

F=3.212

(p > 0.05)

F=25.15

(p < 0.01)

F=10.771

(p < 0.01)

F=11.366

(p < 0.01)

Pn

F=1769.026

(p < 0.01)

F=114.864

(p < 0.01)

F=3185.756

(p < 0.01)

F=6.570

(p < 0.05)

F=46.34

(p < 0.01)

F=140.736

(p < 0.01)

F=39.46

(p < 0.01)

Ci

F=607.036

(p < 0.01)

F=148.467

(p < 0.01)

F=3686.901

(p < 0.01)

F=31.929

(p > 0.05)

F=0.051

(p < 0.01)

F=242.723

(p < 0.01)

F=15.881

(p < 0.01)

Gs

F=2046.216

(p < 0.01)

F=656.639

(p < 0.01)

F=3607.745

(p < 0.01)

F=5.205

(p < 0.05)

F=126.93

(p < 0.01)

F=915.369

(p < 0.01)

F=156.986

(p < 0.01)

Tr

F=10224.121

(p < 0.01)

F=10506.233

(p < 0.01)

F=72188.349

(p < 0.01)

F=444.097

(p < 0.01)

F=20.843

(p < 0.01)

F=5111.278

(p < 0.01)

F=2.361

(p > 0.05)

Membrane characteristics

REC

F=111.259

(p < 0.01)

F=11.732

(p < 0.05)

F=150.98

(p < 0.01)

F=2.978

(p > 0.05)

F=14.421

(p < 0.05)

F=9.006

(p > 0.05)

F=2.38

(p > 0.05)

ROS level

H2O2

F=0.973

(p > 0.05)

F=687.669

(p < 0.01)

F=1734.042

(p < 0.01)

F=1.590

(p > 0.05)

F=3.304

(p > 0.05)

F=254.762

(p < 0.01)

F=0.015

(p > 0.05)

Antioxidant enzymes activity

SOD

F=83.512

(p < 0.01)

F=117.514

(p < 0.01)

F=1093.202

(p < 0.01)

F=2.202

(p > 0.05)

F=1.121

(p > 0.05)

F=83.542

(p < 0.01)

F=42.627

(p < 0.01)

POD

F=40.6

(p < 0.01)

F=5.576

(p < 0.05)

F=96.68

(p < 0.01)

F=0.002

(p > 0.05)

F=2.089

(p > 0.05)

F=2.924

(p > 0.05)

F=0.424

(p > 0.05)

CAT

F=73.32

(p < 0.01)

F=66.286

(p < 0.01)

F=1342.286

(p < 0.01)

F=0.493

(p > 0.05)

F=66.286

(p < 0.01)

F=0.966

(p > 0.05)

F=22.778

(p < 0.01)

APX

F=89.308

(p < 0.01)

F=89.308

(p < 0.01)

F=1231.482

(p < 0.01)

F=5.511

(p < 0.05)

F=64.091

(p < 0.01)

F=39.946

(p < 0.01)

F=0.438

(p > 0.05)

Cell osmotic pressure

RWC

F=71.781

(p < 0.01)

F=13.459

(p < 0.05)

F=120.623

(p < 0.01)

F=5.105

(p < 0.05)

F=11.782

(p < 0.05)

F=10.607

(p < 0.05)

F=4.819

(p < 0.05)

stomatal morphology

PSL

F=2284.202

(p < 0.01)

F=92.877

(p < 0.01)

F=478.062

(p < 0.01)

F=5.864

(p < 0.05)

F=1.873

(p > 0.05)

F=2.111

(p > 0.05)

F=40.174

(p < 0.01)

DSL

F=409.549

(p < 0.01)

F=365.623

(p < 0.01)

F=269.314

(p < 0.01)

F=63.733

(p < 0.01)

F=94.369

(p < 0.01)

F=6.45

(p < 0.05)

F=2.092

(p > 0.05)

PSW

F=7.348

(p < 0.05)

F=32.125

(p < 0.01)

F=71.911

(p < 0.01)

F=0.131

(p > 0.05)

F=0.093

(p > 0.05)

F=4.791

(p < 0.05)

F=0.131

(p > 0.05)

DSW

F=2079.043

(p < 0.01)

F=278.294

(p < 0.01)

F=1382.348

(p < 0.01)

F=6.591

(p < 0.05)

F=2.25

(p > 0.05)

F=83.75

(p < 0.01)

F=46.181

(p < 0.01)

PSA

F=1665.708

(p < 0.01)

F=65.615

(p < 0.01)

F=431.921

(p < 0.01)

F=0.455

(p > 0.05)

F=0.178

(p > 0.05)

F=0.011

(p > 0.05)

F=4.818

(p < 0.05)

DSA

F=1264.801

(p < 0.01)

F=135.369

(p < 0.01)

F=290.6

(p < 0.01)

F=2.98

(p < 0.05)

F=5.316

(p < 0.05)

F=14.257

(p < 0.01)

F=13.409

(p < 0.01)

SW, seedling weight; RW, root weight; SL, seedling length; LA, leaf area; Chl a, chlorophyll a; Chl b, chlorophyll b; Car, carotenoids; Chl a/b, chlorophyll a/b; Pn, net photosynthetic rate; Ci, intercellular CO2 concentration; Gs, stomatal conductance; Tr, transpiration rate; REC, relative electrical conductivity; ROS, reactive oxygen species; H2O2, H2O2 content; SOD, superoxide dismutase activity; POD, peroxidase activity; CAT, catalase activity; APX, ascorbate peroxidase activity; RWC, relative water content; PSL, paraxial surface stomatal length; DSL, dorsal surface stomatal length; PSW, paraxial surface stomatal width; DSW, dorsal surface stomatal width; PSA, paraxial surface stomatal area; DSA, dorsal surface stomatal area.

Moreover, we have described the results of multiple factorial analysis of variance in detail in our manuscript, namely: “Multiple factorial analysis of variance (ANOVA) proved significant for the tested variables (including maize genotype (G), MT (M) application, and Na2CO3 (N) stress) and their physiological and molecular interactions on the 25 tested traits including four growth parameters, eight photosynthetic performances, one membrane characteristics, one ROS levels, four antioxidant enzymes activities, one Cell osmotic pressure, and six stomatal morphology in two maize genotypes seedlings under different treatments (Figures 1-6; Table S1). The findings demonstrated that the resistance to salt stress was controlled by the factor of maize’s own genetic constitution, exogenous MT application, Na2CO3 stress, and their interaction effects. Our results also showed that the genotypes, MT application, and Na2CO3 stress imposed higher influences on these traits than their interactions. We then have re-submitted the manuscript.

Thank you for your consideration.

Open Review: I would not like to sign my review report.

Thanks for your positive comments.

Thank you for your consideration.

Quality of English Language:  English language fine. No issues detected.

Thanks for your positive comments.

Thank you for your consideration.

Does the introduction provide sufficient background and include all relevant references? Can be improved.

Thanks for your positive comments. As suggested, we have revised and improved the introduction section. We then have re-submitted the manuscript.

Thank you for your consideration.

Is the research design appropriate? Must be improved.

Thanks for your positive comments. As suggested, we have improved the experiment design in the manuscript. We then have re-submitted the manuscript.

Thank you for your consideration.

Are the methods adequately described? Can be improved.

Thanks for your positive comments. As suggested, we have improved the descriptions of method in the manuscript in detail. We then have re-submitted the manuscript.

Thank you for your consideration.

Are the results clearly presented? Can be improved.

Thanks for your positive comments. As suggested, we have carefully revised and improved our results section in the manuscript. We then have re-submitted the manuscript.

Thank you for your consideration.

Are the conclusions supported by the results? Can be improved.

Thanks for your positive comments. As suggested, we have improved the conclusion section, namely: “In conclusion, under 100 mM Na2CO3 stress, 150 μM MT application elevated RWC, Pn, Gs, Tr, and Chl content, while reduced REC, Ci, and H2O2 content of leaves in maize seedlings. Meanwhile, the six candidate genes related to photosynthetic pigments and antioxidant homeostasis were up-regulated. They then formed complex interaction relationships to enhance salt tolerance in maize seedlings. Briefly, 150 μM MT application maintains the integrity of photosynthetic organs, scavenging excess ROS, and improving photosynthetic characteristics, subsequently alleviating damage and promoting the growth of maize seedlings under Na2CO3 environment (Figure 9).” In Lines 403-410 on page 13 of the manuscript. We then have re-submitted the manuscript.

Thank you for your consideration.

Sincerely,

Xiaoqiang Zhao professor

State Key Laboratory of Aridland Crop Science, Gansu Agricultural University

E-mail: zhaoxq3324@163.com

2024-9-22

Round 4

Reviewer 2 Report

Comments and Suggestions for Authors

Dear Authors, still lettering is incorrect, if you have applied 3 way ANOVA then how lettering is correct in Fig 1. The value of seedling weight of EMS-104 is about 1.7 g  (Treatment MN) with "b" letter but the value of seedling weight of P-138 is about 1.5 (Treatment M) with "a" letter. How is it possible? Check the lettering in all figures.

Comments on the Quality of English Language

Minor editing of English language required.

Author Response

Thank you for your letter of – and for the referee’s comments concerning our manuscript, “Exogenous Melatonin Reinforces Photosynthesis, Antioxidant Defense, and Their Genes Expression to Ameliorate Na2CO3 Stress in Maize (Manuscript ID: plants-3194664)”. We have carefully studied these comments and have made corresponding corrections to the manuscript, which we describe in detail below. We would like to re-submit the manuscript and that for possible publication on the Special Issue: “Effect of Biotic and Abiotic Factors on the Physiology of Horticultural Plants” of Plants. Thank you very much for your time and consideration.

Reviewer 2

Dear Authors, still lettering is incorrect, if you have applied 3 way ANOVA then how lettering is correct in Fig 1. The value of seedling weight of EMS-104 is about 1.7 g  (Treatment MN) with "b" letter but the value of seedling weight of P-138 is about 1.5 (Treatment M) with "a" letter. How is it possible? Check the lettering in all figures.

Thanks for your positive comments. I am so sorry, I may not have marked the meaning of the letters clearly in all figure legends. The different lowercase letters in Figure 1, 2, 3, 4, 5, 6, and 8 are repreented that “Different lowercase letters among four treatments with single maize genotype indicated significant differences at p < 0.05 level, respectively.” And the “ns, * and **” in Figure 1, 2, 3, 4, 5, 6, and 8 are repreented that “ns: multiple factorial analysis of variance (ANOVA) was nonsignificant at p > 0.05 level; *: multiple factorial ANOVA was significant at p ≤ 0.05 level; **: multiple factorial ANOVA was significant at p ≤ 0.01 level.”.

In these regards, so I have revised the corresponding contents of all figure legends in the manuscript, namely:

Figure 1. Changes of seedling weight (A), root weight (B), seedling length (C), and leaf area (D) of P138 and EMS-104 seedlings under different treatments, including 0 mM Na2CO3+0 μM melatonin (MT) treatment (CK); 100 mM Na2CO3+0 μM MT treatment (N); 0 mM Na2CO3+150 μM MT treatment (M); 100 mM Na2CO3+150 μM MT treatment (MN). Different lowercase letters among four treatments with single maize genotype indicated significant differences at p < 0.05 level, respectively. ns: multiple factorial analysis of variance (ANOVA) was nonsignificant at p > 0.05 level; *: multiple factorial ANOVA was significant at p ≤ 0.05 level; **: multiple factorial ANOVA was significant at p ≤ 0.01 level.

Figure 2. Changes of superoxide dismutase (SOD) activity (A), peroxidase (POD) activity (B), catalase (CAT) activity (C), ascorbate peroxidase (APX) activity (D), and H2O2 content (E) of P138 and EMS-104 seedlings under different treatments, including 0 mM Na2CO3+0 μM melatonin (MT) treatment (CK); 100 mM Na2CO3+0 μM MT treatment (N); 0 mM Na2CO3+150 μM MT treatment (M); 100 mM Na2CO3+150 μM MT treatment (MN). Different lowercase letters among four treatments with single maize genotype indicated significant differences at p < 0.05 level, respectively. ns: multiple factorial analysis of variance (ANOVA) was nonsignificant at p > 0.05 level; *: multiple factorial ANOVA was significant at p 0.05 level; **: multiple factorial ANOVA was significant at p 0.01 level.

Figure 3. Changes of relative electrical conductivity (REC; A), and relative water content (RWC; B) of P138 and EMS-104 seedlings under different treatments, including 0 mM Na2CO3+0 μM melatonin (MT) treatment (CK); 100 mM Na2CO3+0 μM MT treatment (N); 0 mM Na2CO3+150 μM MT treatment (M); 100 mM Na2CO3+150 μM MT treatment (MN). Different lowercase letters among four treatments with single maize genotype indicated significant differences at p < 0.05 level, respectively. ns: multiple factorial analysis of variance (ANOVA) was nonsignificant at p > 0.05 level; *: multiple factorial ANOVA was significant at p ≤ 0.05 level; **: multiple factorial ANOVA was significant at p ≤ 0.01 level.

Figure 4. Changes of net photosynthetic rate (A), stomatal conductance (B), intercellular CO2 concentration (C), and transpiration rate (D) of P138 and EMS-104 seedlings under different treatments, including 0 mM Na2CO3+0 μM melatonin (MT) treatment (CK); 100 mM Na2CO3+0 μM MT treatment (N); 0 mM Na2CO3+150 μM MT treatment (M); 100 mM Na2CO3+150 μM MT treatment (MN). Different lowercase letters among four treatments with single maize genotype indicated significant differences at p < 0.05 level, respectively. ns: multiple factorial analysis of variance (ANOVA) was nonsignificant at p > 0.05 level; *: multiple factorial ANOVA was significant at p ≤ 0.05 level; **: multiple factorial ANOVA was significant at p ≤ 0.01 level.

Figure 5. Changes of chlorophyll a (Chl a; A), chlorophyll b (Chl b; B), carotenoids (Car; C), and chlorophyll a/b (Chl a/b; D) of P138 and EMS-104 seedlings under different treatments, including 0 mM Na2CO3+0 μM melatonin (MT) treatment (CK); 100 mM Na2CO3+0 μM MT treatment (N); 0 mM Na2CO3+150 μM MT treatment (M); 100 mM Na2CO3+150 μM MT treatment (MN). Different lowercase letters among four treatments with single maize genotype indicated significant differences at p < 0.05 level, respectively. ns: multiple factorial analysis of variance (ANOVA) was nonsignificant at p > 0.05 level; *: multiple factorial ANOVA was significant at p ≤ 0.05 level; **: multiple factorial ANOVA was significant at p ≤ 0.01 level.

Figure 6. Changes of paraxial surface stomatal length (A), dorsal surface stomatal length (B), paraxial surface stomatal width (C), dorsal surface stomatal width (D), paraxial surface stomatal area (E), and dorsal surface stomatal area (F) o of P138 and EMS-104 seedlings under different treatments, including 0 mM Na2CO3+0 μM melatonin (MT) treatment (CK); 100 mM Na2CO3+0 μM MT treatment (N); 0 mM Na2CO3+150 μM MT treatment (M); 100 mM Na2CO3+150 μM MT treatment (MN); Different lowercase letters among four treatments with single maize genotype indicated significant differences at p < 0.05 level, respectively. ns: multiple factorial analysis of variance (ANOVA) was nonsignificant at p > 0.05 level; *: multiple factorial ANOVA was significant at p ≤ 0.05 level; **: multiple factorial ANOVA was significant at p ≤ 0.01 level.

Figure 8. Changes of expresion levels of six candiadte genes, including superoxide dismutase (SOD; Zm00001d025106 (A), Zm00001d031908 (B) ), catalase 2 (CAT2; Zm00001d027511) (C), peroxidase 72 (POD72; Zm00001d040364) (D), chlorophyllide a oxygenase chloroplastic (Zm00001d011819) (E), and nine-cis-epoxycarotenoid dioxygenase8 (Zm00001d017766) (F) in leaves of P138 and EMS-104 seedlings under different treatments. 0 mM Na2CO3+0 μM melatonin (MT) treatment (CK); 100 mM Na2CO3+0 μM MT treatment (N); 0 mM Na2CO3+150 μM MT treatment (M); 100 mM Na2CO3+150 μM MT treatment (MN). Different lowercase letters among four treatments with single maize genotype indicated significant differences at p < 0.05 level, respectively.

We then have re-submitted the manuscript.

Thank you for your consideration.

Does the introduction provide sufficient background and include all relevant references? Can be improved.

Thanks for your positive comments. As suggested, I have carefully checked and improved the Introduction and References sections. We then have re-submitted the manuscript.

Thank you for your consideration.

Is the research design appropriate? Must be improved.

Thanks for your positive comments. As suggested, we have improved the experiment design in the manuscript, namely:

4.1. Materials

Previously, a total of 1,041 M3 maize mutant lines were successfully constructed by the elite inbred line P138 pollen from our team, which was treated with 0.5 mg · L-1 ethyl methane sulfonate solution (EMS). P138 is a representative inbred line, derives from the P group, with weak salt tolerance [62]. Our previous study screened out the mutant EMS-104 has stronger salt tolerant than the wild-type P138. Therefore, we conducted a follow-up study on P138 and its mutant EMS-104.

4.2. Plant Growth Conditions.

The seeds of P138 and EMS-104 were sterilized with 0.5% (v/v) sodium hypochlorite (NaClO) solution for 15 min, rinsed five times with ddH2O, and then soaked in 30 mL ddH2O for 24 h at 22±0.5°C environment in darkness. Subsequently the ten soaked seeds were sown in sterilized vermiculite plastic boxes (13 cm × 11 cm), they then cultured in a greenhouse (25±0.5°C temperature; 300 μM · m-2 · s-1 light intensity; 65% relative humidity; 12/12 h light/dark cycle) for 15 d to culture three-leaf seedlings, During the period, the seedlings were washed by 50 mL ddH2O every two days. Next, the three-leaf seedlings were treated four treatments for 7 d, i.e., 0 mM Na2CO3+0 μM MT (ddH2O, CK), 100 mM Na2CO3+0 μM MT(N), 0 mM Na2CO3+150 μM MT(M), and 100 mM Na2CO3+150 μM MT(MN). During the all treatments, the seedlings were also washed by 50 mL above corresponding mixed solution every two days intervals. Each treatment was repeated three times.

 We then have re-submitted the manuscript.

Thank you for your consideration.

Are the methods adequately described? Can be improved.

Thanks for your positive comments. As suggested, we have improved the descriptions of method in the manuscript in detail., namely:

4.1. Materials

Previously, a total of 1,041 M3 maize mutant lines were successfully constructed by the elite inbred line P138 pollen from our team, which was treated with 0.5 mg · L-1 ethyl methane sulfonate solution (EMS). P138 is a representative inbred line, derives from the P group, with weak salt tolerance [62]. Our previous study screened out the mutant EMS-104 has stronger salt tolerant than the wild-type P138. Therefore, we conducted a follow-up study on P138 and its mutant EMS-104.

4.2. Plant Growth Conditions.

The seeds of P138 and EMS-104 were sterilized with 0.5% (v/v) sodium hypochlorite (NaClO) solution for 15 min, rinsed five times with ddH2O, and then soaked in 30 mL ddH2O for 24 h at 22±0.5°C environment in darkness. Subsequently the ten soaked seeds were sown in sterilized vermiculite plastic boxes (13 cm × 11 cm), they then cultured in a greenhouse (25±0.5°C temperature; 300 μM · m-2 · s-1 light intensity; 65% relative humidity; 12/12 h light/dark cycle) for 15 d to culture three-leaf seedlings, During the period, the seedlings were washed by 50 mL ddH2O every two days. Next, the three-leaf seedlings were treated four treatments for 7 d, i.e., 0 mM Na2CO3+0 μM MT (ddH2O, CK), 100 mM Na2CO3+0 μM MT(N), 0 mM Na2CO3+150 μM MT(M), and 100 mM Na2CO3+150 μM MT(MN). During the all treatments, the seedlings were also washed by 50 mL above corresponding mixed solution every two days intervals. Each treatment was repeated three times.

We then have re-submitted the manuscript.

Thank you for your consideration.

Are the results clearly presented? Can be improved.

Thanks for your positive comments. As suggested, we have carefully revised and improved our results section in the manuscript. We then have re-submitted the manuscript.

Thank you for your consideration.

Are the conclusions supported by the results? Can be improved.

Thanks for your positive comments. As suggested, we have further improved the conclusion section, namely: “In conclusion, under 100 mM Na2CO3 stress, 150 μM exogenous applied MT elevated RWC, Pn, Gs, and Tr, and promoted the accumulation of Chl and Car, while reduced REC, Ci, and H2O2 level of leaves in maize seedlings. Meanwhile, the candidate genes related to photosynthetic pigments (Zm00001d011819 and Zm00001d017766) and antioxidant homeostasis (Zm00001d025106, Zm00001d031908, Zm00001d027511, and Zm00001d040364) showed varied up-regulated expression. They then formed complex interaction relationships to enhance salt tolerance in maize seedlings. Briefly, 150 μM MT application maintains the integrity of photosynthetic organs, scavenges excess ROS, and improved photosynthetic characteristics, subsequently alleviates damage and promoted the growth of maize seedlings under Na2CO3 environment (Figure 9). In the manuscript. We then have re-submitted the manuscript.

Thank you for your consideration.

Sincerely,

Xiaoqiang Zhao professor

State Key Laboratory of Aridland Crop Science, Gansu Agricultural University

E-mail: zhaoxq3324@163.com

2024-9-23

Round 5

Reviewer 2 Report

Comments and Suggestions for Authors

Dear Authors, lettering in the figures are still incorrect, if you have applied 3-way ANOVA then how the lettering is correct in Fig 1. The value of seedling weight of EMS-104 is about 1.7 g  (Treatment MN) with "b" letter but the value of seedling weight of P-138 is about 1.5 (Treatment M) with "a" letter. How is it possible? Check the lettering in all figures. Dear authors, when you have performed HSD or LSD test at 0.5% then you will get the treatments with lettering. Need to recheck it because it is not correct.

Correct the statistical analysis statement above the figures you mentioned CRD factorial design with three way anova and also mentioned the test (LSD or HSD or etc.). "Statistical Analysis For all tested traits of the two maize genotypes under all treatments, their one-way ANOVA was performed using the IBM-SPSS Statistics v.20.0 software (SPSS, Chicago, IL., USA; https://www.Ibm.com/products/spss-statistics; accessed on June 16 2024)".

Author Response

Thank you for your letter of – and for the referee’s comments concerning our manuscript, “Exogenous Melatonin Reinforces Photosynthesis, Antioxidant Defense, and Their Genes Expression to Ameliorate Na2CO3 Stress in Maize (Manuscript ID: plants-3194664)”. We have carefully studied these comments and have made corresponding corrections to the manuscript, which we describe in detail below. We would like to re-submit the manuscript and that for possible publication on the Special Issue: “Effect of Biotic and Abiotic Factors on the Physiology of Horticultural Plants” of Plants. Thank you very much for your time and consideration.

Reviewer 2

Dear Authors, lettering in the figures are still incorrect, if you have applied 3-way ANOVA then how the lettering is correct in Fig 1. The value of seedling weight of EMS-104 is about 1.7 g  (Treatment MN) with "b" letter but the value of seedling weight of P-138 is about 1.5 (Treatment M) with "a" letter. How is it possible? Check the lettering in all figures. Dear authors, when you have performed HSD or LSD test at 0.5% then you will get the treatments with lettering. Need to recheck it because it is not correct. Correct the statistical analysis statement above the figures you mentioned CRD factorial design with three way anova and also mentioned the test (LSD or HSD or etc.). "Statistical Analysis For all tested traits of the two maize genotypes under all treatments, their one-way ANOVA was performed using the IBM-SPSS Statistics v.20.0 software (SPSS, Chicago, IL., USA; https://www.Ibm.com/products/spss-statistics; accessed on June 16 2024)".

Thanks for your positive comments. Yes, the all tested traits of the two maize genotypes (P138 and EMS-104) under all treatments were performed one-way ANOVA using the TBM-SPSS Statistics v.20.0 software (SPSS, Chicage, IL., USA).

We have carefully read and considered your positive suggestion, as suggested, we have re-analyzed these all traits in both maize genotypes under all treatments with one –way ANOVA by the IBM-SPSS Statistics v.20.0 software, and have re-provided Figure 1, 2, 3, 4, 5, 6, and 8 in our manuscript, namely:

Figure 1. Changes of seedling weight (A), root weight (B), seedling length (C), and leaf area (D) of P138 and EMS-104 seedlings under different treatments, including 0 mM Na2CO3+0 μM melatonin (MT) treatment (CK); 100 mM Na2CO3+0 μM MT treatment (N); 0 mM Na2CO3+150 μM MT treatment (M); 100 mM Na2CO3+150 μM MT treatment (MN). Different lowercase letters indicated significant differences in p < 0.05 level. ns: multiple factorial analysis of variance (ANOVA) was nonsignificant at p > 0.05 level; *: multiple factorial ANOVA was significant at p ≤ 0.05 level; **: multiple factorial ANOVA was significant at p ≤ 0.01 level.

Figure 2. Changes of superoxide dismutase (SOD) activity (A), peroxidase (POD) activity (B), catalase (CAT) activity (C), ascorbate peroxidase (APX) activity (D), and H2O2 content (E) of P138 and EMS-104 seedlings under different treatments, including 0 mM Na2CO3+0 μM melatonin (MT) treatment (CK); 100 mM Na2CO3+0 μM MT treatment (N); 0 mM Na2CO3+150 μM MT treatment (M); 100 mM Na2CO3+150 μM MT treatment (MN). Different lowercase letters indicated significant differences in p < 0.05 level. ns: multiple factorial analysis of variance (ANOVA) was nonsignificant at p > 0.05 level; *: multiple factorial ANOVA was significant at p ≤ 0.05 level; **: multiple factorial ANOVA was significant at p ≤ 0.01 level.

Figure 3. Changes of relative electrical conductivity (REC; A), and relative water content (RWC; B) of P138 and EMS-104 seedlings under different treatments, including 0 mM Na2CO3+0 μM melatonin (MT) treatment (CK); 100 mM Na2CO3+0 μM MT treatment (N); 0 mM Na2CO3+150 μM MT treatment (M); 100 mM Na2CO3+150 μM MT treatment (MN). Different lowercase letters indicated significant differences in p < 0.05 level. ns: multiple factorial analysis of variance (ANOVA) was nonsignificant at p > 0.05 level; *: multiple factorial ANOVA was significant at p ≤ 0.05 level; **: multiple factorial ANOVA was significant at p ≤ 0.01 level.

Figure 4. Changes of net photosynthetic rate (A), stomatal conductance (B), intercellular CO2 concentration (C), and transpiration rate (D) of P138 and EMS-104 seedlings under different treatments, including 0 mM Na2CO3+0 μM melatonin (MT) treatment (CK); 100 mM Na2CO3+0 μM MT treatment (N); 0 mM Na2CO3+150 μM MT treatment (M); 100 mM Na2CO3+150 μM MT treatment (MN). Different lowercase letters indicated significant differences in p < 0.05 level. ns: multiple factorial analysis of variance (ANOVA) was nonsignificant at p > 0.05 level; *: multiple factorial ANOVA was significant at p ≤ 0.05 level; **: multiple factorial ANOVA was significant at p ≤ 0.01 level.

Figure 5. Changes of chlorophyll a (Chl a; A), chlorophyll b (Chl b; B), carotenoids (Car; C), and chlorophyll a/b (Chl a/b; D) of P138 and EMS-104 seedlings under different treatments, including 0 mM Na2CO3+0 μM melatonin (MT) treatment (CK); 100 mM Na2CO3+0 μM MT treatment (N); 0 mM Na2CO3+150 μM MT treatment (M); 100 mM Na2CO3+150 μM MT treatment (MN). Different lowercase letters indicated significant differences in p < 0.05 level. ns: multiple factorial analysis of variance (ANOVA) was nonsignificant at p > 0.05 level; *: multiple factorial ANOVA was significant at p ≤ 0.05 level; **: multiple factorial ANOVA was significant at p ≤ 0.01 level.

Figure 6. Changes of paraxial surface stomatal length (A), dorsal surface stomatal length (B), paraxial surface stomatal width (C), dorsal surface stomatal width (D), paraxial surface stomatal area (E), and dorsal surface stomatal area (F) o of P138 and EMS-104 seedlings under different treatments, including 0 mM Na2CO3+0 μM melatonin (MT) treatment (CK); 100 mM Na2CO3+0 μM MT treatment (N); 0 mM Na2CO3+150 μM MT treatment (M); 100 mM Na2CO3+150 μM MT treatment (MN); Different lowercase letters indicated significant differences in p < 0.05 level. ns: multiple factorial analysis of variance (ANOVA) was nonsignificant at p > 0.05 level; *: multiple factorial ANOVA was significant at p ≤ 0.05 level; **: multiple factorial ANOVA was significant at p ≤ 0.01 level.

Figure 8. Changes of expresion levels of six candiadte genes, including superoxide dismutase (SOD; Zm00001d025106 (A), Zm00001d031908 (B) ), catalase 2 (CAT2; Zm00001d027511) (C), peroxidase 72 (POD72; Zm00001d040364) (D), chlorophyllide a oxygenase chloroplastic (Zm00001d011819) (E), and nine-cis-epoxycarotenoid dioxygenase8 (Zm00001d017766) (F) in leaves of P138 and EMS-104 seedlings under different treatments. 0 mM Na2CO3+0 μM melatonin (MT) treatment (CK); 100 mM Na2CO3+0 μM MT treatment (N); 0 mM Na2CO3+150 μM MT treatment (M); 100 mM Na2CO3+150 μM MT treatment (MN). Different lowercase letters indicated significant differences in p < 0.05 level.

Moreover, we have carefully revised and improved our manuscript using the “Track Changes” function of the manuscript. We then have re-submitted the manuscript.

Thank you for your consideration.

Open Review: I would not like to sign my review report.

Thanks for your positive comments.

Thank you for your consideration.

Quality of English Language: English language fine. No isuues detected.

Thanks for your positive comments.

Thank you for your consideration.

Does the introduction provide sufficient background and include all relevant references? Yes.

Thanks for your positive comments.

Thank you for your consideration.

Is the research design appropriate? Must be improved.

Thanks for your positive comments. As suggested, according to our experimental aims, we have described our experimental designs and results sections. At the same time, we have re-provided the Figure 1,2, 3, 4,5, 6, and 8. Moreover, we have carefully revised and improved our manuscript using the “Track Changes” function of the manuscript. We then have re-submitted the manuscript.

Thank you for your consideration.

Are the methods adequately described? Yes.

Thanks for your positive comments.

Thank you for your consideration.

Are the results clearly presented? Can be improved.

Thanks for your positive comments. As suggested, we have carefully revised and improved our results section using the “Track Changes” function of the manuscript. We then have re-submitted the manuscript.

Thank you for your consideration.

Are the conclusions supported by the results? Yes.

Thanks for your positive comments.

Thank you for your consideration.

Sincerely,

Xiaoqiang Zhao professor

State Key Laboratory of Aridland Crop Science, Gansu Agricultural University

E-mail: zhaoxq3324@163.com

2024-9-24

Round 6

Reviewer 2 Report

Comments and Suggestions for Authors

Authors have incorporated all the comments. Now, it can be consideration for publication after minor revision.

Results

Please check again lettering of CAT, APX, and stomatal conductance parameters.

Statistical analysis:

"their one-way ANOVA was performed" should replaced with "their three-way ANOVA was performed"